# Group Meritocratic Fairness in Linear Contextual Bandits

**Riccardo Grazzi**[1,2]*, **Arya Akhavan**[1,3], **John Isak Texas Falk**[1,2],
**Leonardo Cella**[1], **Massimiliano Pontil**[1,2]

[1]CSML, Istituto Italiano di Tecnologia, Genoa, Italy
[2]Dept. of Computer Science, University College London, UK
[3]CREST, ENSAE, Institut Polytechnique de Paris, France

## Abstract

We study the linear contextual bandit problem where an agent has to select one candidate from a pool and each candidate belongs to a sensitive group. In this setting, candidates' rewards may not be directly comparable between groups, for example when the agent is an employer hiring candidates from different ethnic groups and some groups have a lower reward due to discriminatory bias and/or social injustice. We propose a notion of fairness that states that the agent's policy is fair when it selects a candidate with highest relative rank, which measures how good the reward is when compared to candidates from the same group. This is a very strong notion of fairness, since the relative rank is not directly observed by the agent and depends on the underlying reward model and on the distribution of rewards. Thus we study the problem of learning a policy which approximates a fair policy under the condition that the contexts are independent between groups and the distribution of rewards of each group is absolutely continuous. In particular, we design a greedy policy which at each round constructs a ridge regression estimate from the observed context-reward pairs, and then computes an estimate of the relative rank of each candidate using the empirical cumulative distribution function. We prove that, despite its simplicity and the lack of an initial exploration phase, the greedy policy achieves, up to log factors and with high probability, a fair pseudo-regret of order $\sqrt{dT}$ after $T$ rounds, where $d$ is the dimension of the context vectors. The policy also satisfies demographic parity at each round when averaged over all possible information available before the selection. Finally, we use simulated settings and experiments on the US census data to show that our policy achieves sub-linear fair pseudo-regret also in practice.

## 1 Introduction

Consider a sequential decision making problem where at each round an employer has to select one candidate from a pool to hire for a job. The employer does not know how well a candidate will perform if hired, but they can learn it over time by measuring the performance of previously selected similar candidates. This scenario can be formalized as a (linear) contextual bandit problem (see [2, 12, 26] and references therein), where each candidate is represented by a context vector, and after the employer (or agent) chooses a candidate, it receives a reward, i.e. a scalar value measuring the true performance of the candidate, which depends (linearly) on the context.

In the above framework, the typical objective is to find a policy for the employer to select candidates with the highest rewards [1–3, 26]. However, in some important scenarios this objective may not

---
*riccardo.grazzi@iit.it

36th Conference on Neural Information Processing Systems (NeurIPS 2022).

be appropriate; if candidates belong to different sensitive groups (e.g. based on ethnicity, gender, etc.) the resulting policy might discriminate or even exclude some groups completely in the selection process. This may happen when some groups have lower expected reward than others, e.g. because they acquired less skills due lower financial support. Another example arises when each candidate in the pool, if selected, will perform a different kind of job, and the associated reward is job-specific. For instance, if the employer is a university and each candidate is a researcher in a different discipline, then the rewards associated to different disciplines will be substantially different and incomparable, e.g. citation counts vary greatly among different subjects; see [23] for a discussion. In both of the above scenarios, it is unfair to directly compare rewards of candidates belonging to different groups.

A simple way to deal with this issue would be to select the candidate to hire uniformly at random. This policy satisfies a notion of fairness called *demographic parity* (see [7, 31] and references therein), which requires the probability of selecting a candidate from a given group to be equal for all groups. However, as is apparent, this approach completely ignores the employer's goal of selecting good candidates and is also unfair to candidates who spent effort acquiring credentials for the job. In this work, we provide a fair way of comparing candidates from different groups via the *relative rank*, that is the cumulative distribution function (CDF) value of the reward of the candidate where the distribution is that of the rewards of the candidate's group. We call a policy *group meritocratic fair* (GMF) if it always selects a candidate with the highest relative rank. Such a policy is meritocratic but only in terms of the within-group performance. A closely related idea has been introduced in [23] for settings where the candidates' rewards are available before the selection, while we are not aware of a similar notion in the multi-armed bandits literature.

A GMF policy requires the knowledge of the relative rank of each candidate which is not directly observed by the agent and depends on the underlying reward model and on the distributions of rewards. Moreover, to estimate the relative rank from the observed rewards and contexts it is necessary to learn the CDF of the rewards of each group, which adds a challenge to the standard linear contextual bandit framework where only the linear relation between contexts and rewards has to be learned. Due to this, a learned policy cannot be GMF at all rounds, thus we study the problem of learning a policy which minimizes the *fair regret*, that is the cumulative difference between the relative rank of the candidate chosen by a GMF policy and the candidate chosen by a learning policy.

For this purpose, we design a greedy policy, which at each round uses the following two-stage strategy. Firstly, it constructs a ridge regression estimate which maps contexts to rewards. Secondly, it computes an estimate of the relative rank of each candidate using the empirical CDF of the estimated rewards. We show that the proposed policy achieves, under some reasonable assumptions and after $T$ rounds, $\tilde{O}(K^3 + \sqrt{dT})$ fair regret with high probability, where $d$ is the dimension of the context vectors and $K$ is the number of candidates in the pool. Notably, our policy does not require an initial exploration phase and satisfies demographic parity at each round when averaged over all possible random draws of the information avaliable to the agent before the decision, i.e. current contexts and previously received contexts, actions and rewards.

**Contributions and Organization.** After a review of previous work in Sec. 2, we introduce the learning problem and the proposed fairness notion in Sec. 3. To simplify the exposition, we assume that each arm corresponds to a sensitive group. In Sec. 4 we propose a greedy policy which jointly learns the underlying regression model and the CDF of each group. We derive a $\tilde{O}(\sqrt{dT})$ regret bound for our policy in Sec. 5. In Sec. 6 we present an illustrative simulation experiment with diverse reward distributions. In Sec. 7, we extend our policy and results to the case where candidates from the same arm can belong to different groups and show the efficacy of our approach with an experiment on the US census data where the sensitive group (ethnicity) is drawn at random together with the context. We draw conclusions in Sec. 8. Code at `https://github.com/CSML-IIT-UCL/GMFbandits`

**Notation.** We use $\langle \cdot, \cdot \rangle$ for the scalar product. For $K \in \mathbb{N}$ we have $[K] = \{1, \ldots, K\}$. Let $\psi$ be a scalar random variable, for each $a \in [K]$ and $\mu^* \in \mathbb{R}^d$, we denote with $\mathcal{F}_\psi$, $\mathcal{F}_a$ the CDF of $\psi$ and $\langle \mu^*, X_a \rangle$ respectively. If $X$ is a continuous random vector with values in $\mathbb{R}^d$, we denote with $f_X : \mathbb{R}^d \to [0, \infty)$ its probability density function. For any $s \in \mathbb{N}$, we denote $\mathbb{I}_s$ as the $s \times s$ identity matrix. For a random variable $Y \in \mathbb{R}^s$, we call $Y$ an absolutely continuous random variable, if its distribution is an absolutely continuous measure with respect to the Lebesgue measure on $\mathbb{R}^s$. For a positive semi-definite matrix $D$, we denote $\lambda_{\min}(D)$ and $\lambda_{\min}^+(D)$ the smallest eigenvalue of $D$, and the smallest non-zero eigenvalue of $D$ respectively. $\mathrm{Supp}(X)$ indicates the support of a random variable $X$. We also denote with $\mathcal{U}[S]$, the uniform distribution over the set $S$.

## 2 Related Works

In recent years algorithmic fairness has received a lot of attention, becoming a large area of machine learning research. The potential for learning algorithms to amplify pre-existing bias and cause harm to human beings has triggered researchers to study solutions to mitigate or remove unfairness of the learned predictor, see [4, 8, 11, 14, 16, 19, 20, 22, 24, 25, 29, 35–37, 39] and references therein. Fairness in sequential decision problems (see [38] for a survey) is usually divided into two categories: group fairness (GF) and individual fairness. We give an overview of these notions below.

GF requires some statistical measure to be (approximately) equal across different sensitive groups. A prominent example relevant to this work is *demographic parity*, which requires that the probability that the policy selects a candidate from a given group should be the same for all groups. A similar notion is used by [10, 32], where the probability that the policy selects a candidate has to always be greater than a given threshold for all candidates. [27] impose a weaker requirement concerning the expected fraction of candidates selected from each group. Other examples of GF in sequential decision problems are *equal opportunity* [5] and *equalized odds* [6]. Under some assumptions on the distributions of the contexts, our GMF and greedy policies satisfy variants of demographic parity at each round.

Individual fairness can be divided in two categories: fairness through awareness (FA) [28, 34] and meritocratic fairness (MF) [21, 22]. FA is based on the idea that similar individuals should be treated similarly and is designed to avoid "winner takes all" scenarios where some individuals cannot be selected when they have a lower reward than others in the pool, even if the difference between rewards is very small. For example, [34] propose a policy where the probability of selecting a context over another is lower when the context has a lower reward, but is never zero. MF instead requires that less qualified individual should not be favored over more qualified ones, which could happen during the learning process. For example [22] proposes an algorithm where the policy selects the arm uniformly at random among the best arms with overlapping confidence intervals. This guarantees meritocratic fairness at each round but comes at a cost in terms of regret.

Our definition of fairness falls between group and meritocratic fairness. It is meritocratic because it states that a candidate with a worse relative rank than another should never be selected. It is also based on groups since the relative ranks directly depend on the distribution of rewards of each group. A similar idea of fairness based on relative rank has been introduced in [23], which study the problem of selecting candidates from different groups based on their scalar-valued score when the scores between groups are incomparable (e.g. number of citations in different research areas). Contrary to our work, where the (noisy) rewards are observed only for the selected candidates, in [23] the noiseless scores for all candidates can be accessed before the selection. This difference makes the estimation of the relative rank simpler in [23], as the rewards CDFs can be estimated more efficiently.

## 3 Group Meritocratic Fairness in Linear Contextual Bandits

We consider the linear contextual bandit setup [2] where at each round $t \in [T]$, an agent receives a set of feature vectors $\{X_{t,a}\}_{a=1}^{K}$ with $X_{t,a} \subset \mathbb{R}^d$ sampled from the environment, one for each arm $a \in [K]$. We assume that context (or candidate) $X_{t,a}$ has an associated reward $\langle \mu^*, X_{t,a} \rangle$ where $\mu^* \in \mathbb{R}^d$ is unknown to the agent. After the agent selects the arm $a_t$, it receives the noisy reward equal to $r_{t,a_t} = \langle \mu^*, X_{t,a_t} \rangle + \eta_t$, where $\eta_t$ is some scalar noise (formally specified later). In addition, we assume that each arm represents a fixed sensitive group (e.g. based on ethnicity, gender, etc.). The latter assumption simplifies the presentation but implies that at each round the agent receives exactly one candidate for each group. This can be too restrictive e.g. when candidates are sampled i.i.d. together with their group and/or some groups are minorities. However, our results can be easily adapted to more realistic settings without such assumption, as we show in Sec. 7 and more rigorously in Appendix E. Excluding these sections, we use arm and group interchangeably in all that follows.

Usually, the goal of the agent is to maximise the expected cumulative reward $\sum_{t=1}^{T} \langle \mu^*, X_{t,a_t} \rangle$. Since as we previously explained, this objective might be unfair to some of the sensitive groups, we instead use a different kind of reward which measures the relative performance of a candidate compared to others of the same arm/group. First, we additionally assume, for each group $a$, that $\{X_{t,a}\}_{t=1}^{T}$ are i.i.d and have the same distribution of $X_a$, which we define to be a random variable with unknown distribution. We call the distribution of $\langle \mu^*, X_a \rangle$ the reward distribution of arm $a$ and denote with $\mathcal{F}_a$

its CDF, i.e. $\mathcal{F}_a(r) = \mathbb{P}(\langle \mu^*, X_a \rangle \leq r)$ for every $r \in \mathbb{R}$. Then, we introduce the *relative rank* of candidate $X_{t,a}$ as $\mathcal{F}_a(\langle \mu^*, X_{t,a} \rangle)$, that is the probability that a sample from the reward distribution of arm $a$ is lower than the reward of $X_{t,a}$. We argue that the relative rank, allows to have a fair way of comparing candidates from different groups and introduce the following fairness definition.

**Definition 3.1** (Group Meritocratic Fairness). *A policy $\{a_t^*\}_{t=1}^{\infty}$ is group meritocratic fair (GMF) if for all $t \in \mathbb{N}, a \in [K]$*

$$\mathcal{F}_{a_t^*}(\langle \mu^*, X_{t,a_t^*} \rangle) \geq \mathcal{F}_a(\langle \mu^*, X_{t,a} \rangle) \ .$$

A GMF policy chooses candidates with the highest reward compared to candidates from the same group. This is a strong definition of fairness which is impossible to satisfy at each round for a learned policy. As in standard linear contextual bandits, $\mu^*$ is unknown and must be learned. In this setting however, we have the additional challenge of learning the CDF for the rewards of each arm, $\mathcal{F}_a$. Thus, we will focus on how to learn a GMF policy by introducing the following regret definition.

**Definition 3.2** (Fair Pseudo-Regret). *Let $T \in \mathbb{N}$, $\{a_t\}_{t=1}^{T}$ be the evaluated policy and $\{a_t^*\}_{t=1}^{T}$ be a GMF policy. Then we denote by (cumulative) fair pseudo-regret the quantity*

$$R_F(T) := \sum_{t=1}^{T} \mathcal{F}_{a_t^*}(\langle \mu^*, X_{t,a_t^*} \rangle) - \mathcal{F}_{a_t}(\langle \mu^*, X_{t,a_t} \rangle) \ . \tag{1}$$

The goal of the learned policy will be to minimize the fair pseudo-regret, since a policy with sublinear fair pseudo-regret will get closer and closer to a GMF fair policy over time.

**Remark 3.1.** *The fair pseudo-regret resembles the standard pseudo-regret defined as*

$$R(T) := \sum_{t=1}^{T} \langle \mu^*, X_{t,a_t^{\mathrm{opt}}} \rangle - \langle \mu^*, X_{t,a_t} \rangle \quad \text{with} \quad a_t^{\mathrm{opt}} \in \underset{a \in [K]}{\mathrm{argmax}} \langle \mu^*, X_{t,a} \rangle \ ,$$

*where rewards are replaced by relative ranks and $a_t^{\mathrm{opt}}$ by the GMF policy $a_t^*$. Furthermore, since the CDF restricted to the support is strictly increasing, when the reward distributions are the same for each arm, i.e. $\mathcal{F}_a = \mathcal{F}_{a'}$ for all $a, a' \in [K]$, then a policy minimizing the fair pseudo-regret also minimizes the standard pseudo-regret and vice versa. This is not true in the general case, where fair and standard pseudo-regrets are often competing objectives. For example, when $\{\langle \mu^*, X_a \rangle\}_{a=1}^{K}$ are independent and absolutely continuous and there exists $\hat{a}$ such that $\langle \mu^*, X_{\hat{a}} \rangle > \langle \mu^*, X_a \rangle$ for every $a \neq \hat{a}$, then for every $t$, $a_t^{\mathrm{opt}} = \hat{a}$, while as we will show in Proposition 3.1, $a_t^*$ selects each arm with equal probability. Thus, with non-zero probability $a_t^{\mathrm{opt}}$ has a linear fair pseudo-regret while $a_t^*$ has a linear standard pseudo-regret. Moreover, in Appendix F, for $K = 2$, we show that if $\langle \mu^*, X_1 \rangle$ and $\langle \mu^*, X_2 \rangle$ are independent, absolutely continuous, but not identically distributed, then the GMF policy has a linear standard regret and $\{a_t^{opt}\}_{t=1}^{\infty}$ has a linear fair regret with positive probability.*

Learning a GMF policy brings several challenges. The relative rank is not directly observed by the agent, which receives instead only the noisy reward. This implies that the agent has to estimate $\mathcal{F}_a$, which in general might not even be Lipschitz continuous. This is the main reason why we restrict our analysis to the case where the rewards $\{\langle \mu^*, X_a \rangle\}_{a=1}^{K}$ are independent and absolutely continuous. In particular, for any $t \geq 0$, let $\mathcal{H}_t^- := \cup_{i=1}^{t} \left\{ \{X_{i,a}\}_{a=1}^{K}, r_{i,a_i}, a_i \right\}$ with $\mathcal{H}_0^- = \varnothing$ and $\mathcal{H}_t := \mathcal{H}_t^- \cup \{ \{X_{t+1,a}\}_{a=1}^{K} \}$ be respectively the history and the information available for the decision at round $t + 1$, then the following holds.

**Proposition 3.1** (GMF policy satisfies *history-agnostic demographic parity*). *Let $\{\langle \mu^*, X_a \rangle\}_{a=1}^{K}$ be independent and absolutely continuous and for every $a \in [K], t \in \mathbb{N}$, let $X_{t,a}$ be an i.i.d. copy of $X_a$. Then for every $t \in \mathbb{N}$, $\{\mathcal{F}_a(\langle \mu^*, X_{t,a} \rangle)\}_{a=1}^{K}$ are i.i.d. uniform on $[0,1]$ and*

$$\mathbb{P}(a_t^* = a \mid \mathcal{H}_{t-1}^-) = \frac{1}{K} \qquad \forall a \in [K], \tag{2}$$

*for any GMF policy $\{a_t^*\}_{t=1}^{\infty}$. Note, the randomness lies exclusively in the current contexts $\{X_{t,a}\}_{a=1}^{K}$.*

*Proof.* Let $\psi_a := \mathcal{F}_a(\langle \mu^*, X_{t,a} \rangle)$. From the assumptions $\{\psi_a\}_{a=1}^{K}$ are i.i.d random variables, independent from $\mathcal{H}_{t-1}^-$, with uniform distribution on $[0,1]$ (see [9, Theorem 2.1.10]). Hence $\forall a_1, a_2 \in [K]$: $\mathbb{P}(\psi_{a_1} = \psi_{a_2}) = 0, \mathbb{P}(a_t^* = a \mid \mathcal{H}_{t-1}^-) = \mathbb{P}(a_t^* = a)$ and

$$\mathbb{P}(a_t^* = a_1) = \mathbb{P}(\psi_{a_1} > \psi_{a'}, \forall a' \neq a_1) = \mathbb{P}(\psi_{a_2} > \psi_{a'}, \forall a' \neq a_2) = \mathbb{P}(a_t^* = a_2) = 1/K \ .$$

$\square$

We call property (2) history-agnostic demographic parity since it states that, at each round, the policy selects all groups with equal probability regardless of the history. Recall that in our setup each arm corresponds to a sensitive group. Proposition 3.1 ensures that a GMF policy will keep exploring regardless of the history. This fact plays a key role in the design of our policy, which is greedy without the need of an exploration phase.

**Remark 3.2.** *Note that in the standard linear contextual bandit setting, the optimal policy $a_t^{\mathrm{opt}}$ does not necessarily satisfy (2) even when we assume that $\{\langle \mu^*, X_a \rangle\}_{a=1}^K$ are independent and absolutely continuous. This is true since when the rewards of one arm are always lower than at least one of the other arms, that arm will never be selected by the optimal policy.*

In the following, we state and discuss the assumptions made for the analysis of our greedy policy.

**Assumption A.** *Let $\mu^* \in \mathbb{R}^d$ be the underlying reward model. We assume that:*

(i) *The noise random variable $\eta_t$ is zero mean $R$-subgaussian, conditioned on $\mathcal{H}_{t-1}$.*

(ii) *Let $X_a$ be a random variable with values in $\mathbb{R}^d$ and such that $\|X_a\|_2 \leq L$ almost surely. For any $a \in [K]$, $\{X_{i,a}\}_{i=1}^T$ are i.i.d. copies of $X_a$.*

(iii) *The random variables $\{X_a\}_{a=1}^K$ are mutually independent.*

(iv) *For every $a \in [K]$, there exist $d_a \geq 1$, an absolutely continuous random variable $Y_a$ with values in $\mathbb{R}^{d_a}$ admitting a density $f_a$, $B_a \in \mathbb{R}^{d \times d_a}$ and $c_a \in \mathbb{R}^d$ such that $B_a^\top B_a = \mathbb{I}_{d_a}$,*

$$X_a = B_a Y_a + c_a \quad and \quad \mu^{*\top} B_a \neq 0 \ .$$

Assumption A(i) is a standard assumption on the noise in stochastic bandits. A(ii) implies that the actions taken by the policy do not affect future contexts. This is needed to allow the learning of the distribution of rewards for each group and is also used in [10, 27]. A(iv) implies that $\langle \mu^*, X_a \rangle$ is absolutely continuous and is satisfied when $X_a$ is absolutely continuous in a subspace of $\mathbb{R}^d$ which is not orthogonal to $\mu^*$ [2]. This fact combined with A(iii) ensures that Proposition 3.1 holds. Assumptions A(iii)-(iv) are specific to our setting and a current limitation of the analysis. Notice however, that A(iii) is reasonable when the groups are sufficiently isolated, e.g. each context is sourced from a different country/group, while assuming that the rewards $\langle \mu^*, X_a \rangle$ are absolutely continuous is natural when the contexts contain continuous attributes. Furthermore A(iv) allows $\mu^*$ to act differently on each group, similarly to the case when there is a different reward vector for each sensitive group. An example of this is showed in the simulation experiment in Sec. 6.

## 4 The Fair-Greedy Policy

If Proposition 3.1 holds, then there is no arm with relative rank always strictly worse than the others and any learned policy with sub-linear fair pseudo-regret will select all arms with equal probability in the limit when the number of rounds goes to infinity. Hence, using confidence intervals will not help in decreasing the probability that one arm is selected. Furthermore, estimating the relative ranks $\{\mathcal{F}_a(\langle \mu^*, X_{t,a} \rangle)\}_{a=1}^K$ is challenging, since they are not directly observed and using the past noisy rewards $\{r_{i,a_i}\}_{i=1}^{t-1}$ to construct the empirical CDF for each group, similarly to [23], can be inaccurate due to the presence of noise.

For the reasons above, we propose the greedy approach in Alg. 1, which uses the following two-stage procedure at each round $t$. First it assembles the previously selected contexts and corresponding rewards from iterate 1 up to $\tilde{t} = \lfloor (t-1)/2 \rfloor$ (line 4) in order to construct an estimate $\mu_{\tilde{t}}$ of $\mu^*$ (line 5), which is a noisy version of the ridge regression estimate. Secondly, for each arm $a$, our policy computes an estimate of the relative rank $\mathcal{F}_a(\langle \mu^*, X_{t,a} \rangle)$, namely $\hat{\mathcal{F}}_{t,a}(\langle \mu_{\tilde{t}}, X_{t,a} \rangle)$, which is the empirical CDF value of $\langle \mu_{\tilde{t}}, X_{t,a} \rangle$ and is constructed using $\mu_{\tilde{t}}$ and the contexts from round $\tilde{t} + 1$ up to $t$ (line 6). Lastly, it selects $a_t$ uniformly at random among the arms maximizing the relative rank estimate (line 7).

---

[2]E.g. $X_a$ cannot be sum of random variables that are independent and absolutely continuous in orthogonal subspaces of $\mathbb{R}^d$.

---

**Algorithm 1** Fair-Greedy

---

1: **Requires** regularization parameter $\lambda > 0$ and noise magnitude $\rho \in (0, 1]$ .
2: **for** $t = 1 \dots T$ **do**
3:     Receive contexts $\{X_{t,a}\}_{a=1}^{K}$
4:     Set $\tilde{t} = \lfloor (t-1)/2 \rfloor$, $X_{1:\tilde{t}} = (X_{1,a_1}, \dots, X_{\tilde{t},a_{\tilde{t}}})^{\top}$, $r_{1:\tilde{t}} = (r_{1,a_1}, \dots, r_{\tilde{t},a_{\tilde{t}}})$.
5:     **If** $\tilde{t} = 0$ set $\mu_{\tilde{t}} = 0$, **else** let $V_{\tilde{t}} := X_{1:\tilde{t}}^{\top} X_{1:\tilde{t}} + \lambda \mathbb{I}_d$, generate $\gamma_{\tilde{t}} \sim \mathcal{N}(0, \mathbb{I}_d)$ and compute

$$\mu_{\tilde{t}} := V_{\tilde{t}}^{-1} X_{1:\tilde{t}}^{\top} r_{1:\tilde{t}} + \frac{\rho}{d\sqrt{\tilde{t}}} \cdot \gamma_{\tilde{t}} \ .$$

6:     For each $a \in [K]$ compute

$$\hat{\mathcal{F}}_{t,a}(\langle \mu_{\tilde{t}}, X_{t,a} \rangle) := (t - 1 - \tilde{t})^{-1} \sum_{s=\tilde{t}+1}^{t-1} \mathbb{1}\left\{ \langle \mu_{\tilde{t}}, X_{s,a} \rangle \leq \langle \mu_{\tilde{t}}, X_{t,a} \rangle \right\} \ .$$

7:     Sample action

$$a_t \sim \mathcal{U}\left[ \operatorname*{argmax}_{a \in [K]} \hat{\mathcal{F}}_{t,a}(\langle \mu_{\tilde{t}}, X_{t,a} \rangle) \right] \ .$$

8:     Observe noisy reward $r_{t,a_t} = \langle \mu, X_{t,a_t} \rangle + \eta_t$.
9: **end for**

---

Fair-Greedy has two hyperparameters $\lambda$ and $\rho$, although the latter can be set arbitrarily small without affecting the regret. Moreover, it is greedy as at each time $t$, it always selects from the arms the one with the highest currently estimated relative rank. However, contrary to standard greedy approaches in bandits, Fair-Greedy does not require an initial exploration phase because it naturally explores all arms, as the following lemma and remark show.

**Lemma 4.1** (Fair-Greedy satisfies *information averaged demographic parity*)**.** *Let $a_t$ be the action taken by Fair-Greedy at time $t$ and let Assumption A be satisfied. Then, for all $t \geq 1$ we have*

$$\mathbb{P}(a_t = a) = \frac{1}{K} \ . \tag{3}$$

*Proof sketch (proof in Appendix B).* The noise term in $\mu_{\tilde{t}}$ ensures that $\mu_{\tilde{t}}$ is absolutely continuous and hence $\mu_{\tilde{t}}^{\top} B_a \neq 0$ almost surely. Combining this with Assumption A(iv) we obtain that $\langle \mu_{\tilde{t}}, X_a \rangle$ is also absolutely continuous (see Lemma A.1). Moreover, thanks to Assumption A(ii)(iii) we can show that the random variables in $\{\hat{\mathcal{F}}_{t,a}(\langle \mu_{\tilde{t}}, X_{t,a} \rangle)\}_{a=1}^{K}$ are i.i.d. when conditioned on $\mu_{\tilde{t}}$. Note that $a_t$ is sampled uniformly form the argmax of i.i.d. random variables, when conditioned on $\mu_{\tilde{t}}$, which implies $\mathbb{P}(a_t = a \,|\, \mu_{\tilde{t}}) = 1/K$. The statement follows by taking the expectation over $\mu_{\tilde{t}}$. $\qquad\square$

**Remark 4.1.** *It is easy to verify (through Lemma 4.1) that at any number of rounds $T$, the Fair-Greedy policy selects in expectation $T/K$ candidates from every group, i.e. $\mathbb{E}\left[ \sum_{t=1}^{T} \mathbb{1}\{a_t = a\} \right] = \frac{T}{K}$ for every $a \in [K]$. This also holds for the GMF policy and the one selecting arms uniformly at random.*

Since $\mathbb{P}(a_t = a) = \mathbb{E}_{\mathcal{H}_{t-1}}[\mathbb{P}(a_t = a \,|\, \mathcal{H}_{t-1})]$, with $\mathcal{H}_{t-1}$ being the information available to the policy before making a decision at round $t$, we call the property in (3) information-averaged demographic parity, which is weaker than history-agnostic demographic parity (in (2)). However, our analysis still requires a lower bound on $\mathbb{P}(a_t = a \,|\, \mathcal{H}_{t-1}^-)$ which is presented in the next section.

**Remark 4.2** (Computational cost of Fair-Greedy)**.** *Compared to common linear contextual bandits approaches based on ridge regression, Alg. 1 has an higher computational and memory cost which grow linearly with $t$. $\mu_{\tilde{t}}$ requires us to compute the product of $V_{\tilde{t}}^{-1}$ and $X_{1:\tilde{t}}^{\top} r_{1:\tilde{t}}$, which can be stored using $d^2$ and $d$ values respectively and updated online (via sherman-morrison [17]). However, Alg. 1 also requires, at each round $t$, to keep in memory $K(t - 1 - \tilde{t})$ $d$-dimensional contexts and to compute the same number of scalar products to construct the empirical CDF for all $K$ groups.*

# 5 Regret Analysis

In this section we present the analysis leading to the high probability $\tilde{O}(K^3 + \sqrt{dT})$ upper bound on the fair pseudo-regret of the greedy policy in Alg. 1. We start by showing two key properties of CDF functions in the following lemma (proof in Appendix C.1). Recall that for a continuous random variable $Z$ we denote by $f_Z$ the associated probability density function (PDF).

**Lemma 5.1.** *Let Assumption A(iv) hold and set $\forall a \in [K]$, $Z_a := \langle \mu^*, X_a \rangle$ so that $\mathcal{F}_a = \mathcal{F}_{Z_a}$ and $M := \max_{a \in [K], z \in \mathbb{R}} f_{Z_a}(z) < +\infty$ as the maximum PDF value of the rewards of all groups. Then, the following two statements are true.*

(i) *$\mathcal{F}_a$ is Lipschitz continuous for every $a \in [K]$, and in particular for any $r, r' \in \mathbb{R}$ we have*
$$\sup_{a \in [K]} |\mathcal{F}_a(r) - \mathcal{F}_a(r')| \leq M|r - r'| \ .$$

(ii) *For every $a \in [K]$, let $\mu \in \mathbb{R}^d$, $\tilde{Z}_a := \langle \mu, X_a \rangle$. Then we have*
$$\sup_{a \in [K], r \in \mathbb{R}} |\mathcal{F}_a(r) - \mathcal{F}_{\tilde{Z}_a}(r)| \leq 2M\|\mu^* - \mu\|\|x_{\max}\|_* \ ,$$

*for any norm $\|\cdot\|$ with dual norm $\|\cdot\|_*$ , where $\|x_{\max}\|_* := \sup_{x \in \cup_{a=1}^K \mathrm{Supp}(X_a)} \|x\|_*$ and $\mathrm{Supp}(X_a)$ is the support of the random variable $X_a$ .*

Lemma 5.1(i) bounds the Lipschitz constant of $\mathcal{F}_a$ and its derivation is straightforward. Lemma 5.1(ii) is needed since we only have access to an estimate of $\mu^*$, which will take the role of $\mu$. Its derivation is more subtle and could be of independent interest. By using Lemma 5.1 and the Dvoretzky–Kiefer–Wolfowitz-Massart (DKWM) inequality [15, 30] to bound the gap between CDF and empirical CDF, we obtain the following result.

**Lemma 5.2** (Instant regret bound). *Let Assumption A(ii)(iv) hold and $a_t$ to be generated by Alg. 1. Then with probability at least $1 - \delta/4$, for all $t$ such that $3 \leq t \leq T$ we have*

$$\mathcal{F}_{a_t^*}(\langle \mu^*, X_{t, a_t^*}\rangle) - \mathcal{F}_{a_t}(\langle \mu^*, X_{t, a_t}\rangle) \leq 6M\|\mu^* - \mu_{\tilde{t}}\|_{V_{\tilde{t}}}\|x_{\max}\|_{V_{\tilde{t}}^{-1}} + 2\sqrt{\frac{\log(8KT/\delta)}{t - 1}} \ ,$$

*where $\|x_{\max}\|_{V_{\tilde{t}}^{-1}} := \sup_{x \in \cup_{a=1}^K \mathrm{Supp}(X_a)} \|x\|_{V_{\tilde{t}}^{-1}}$.*

*Proof.* Let $Z_t := \langle \mu_{\tilde{t}}, X_{a_t}\rangle$, $Z_t^* := \langle \mu_{\tilde{t}}, X_{a_t^*}\rangle$ and $\mathcal{F}_{Z_t}, \mathcal{F}_{Z_t^*}$ be their CDF conditioned on $\mu_{\tilde{t}}$, $a_t$, and $a_t^*$. Let also $R_{\mathrm{inst}}(t) := \mathcal{F}_{a_t^*}(\langle \mu^*, X_{t, a_t^*}\rangle) - \mathcal{F}_{a_t}(\langle \mu^*, X_{t, a_t}\rangle)$ . Then we can write

$$R_{\mathrm{inst}}(t) = \underbrace{\mathcal{F}_{a_t^*}(\langle \mu^*, X_{t, a_t^*}\rangle) - \mathcal{F}_{a_t^*}(\langle \mu_{\tilde{t}}, X_{t, a_t^*}\rangle)}_{(I)} + \underbrace{\mathcal{F}_{a_t^*}(\langle \mu_{\tilde{t}}, X_{t, a_t^*}\rangle) - \mathcal{F}_{Z_t^*}(\langle \mu_{\tilde{t}}, X_{t, a_t^*}\rangle)}_{(II)}$$

$$+ \underbrace{\mathcal{F}_{Z_t^*}(\langle \mu_{\tilde{t}}, X_{t, a_t^*}\rangle) - \hat{\mathcal{F}}_{t, a_t^*}(\langle \mu_{\tilde{t}}, X_{t, a_t^*}\rangle)}_{(III)} + \underbrace{\hat{\mathcal{F}}_{t, a_t^*}(\langle \mu_{\tilde{t}}, X_{t, a_t^*}\rangle) - \hat{\mathcal{F}}_{t, a_t}(\langle \mu_{\tilde{t}}, X_{t, a_t}\rangle)}_{(IV)}$$

$$+ \underbrace{\hat{\mathcal{F}}_{t, a_t}(\langle \mu_{\tilde{t}}, X_{t, a_t}\rangle) - \mathcal{F}_{Z_t}(\langle \mu_{\tilde{t}}, X_{t, a_t}\rangle)}_{(V)} + \underbrace{\mathcal{F}_{Z_t}(\langle \mu_{\tilde{t}}, X_{t, a_t}\rangle) - \mathcal{F}_{a_t}(\langle \mu_{\tilde{t}}, X_{t, a_t}\rangle)}_{(VI)}$$

$$+ \underbrace{\mathcal{F}_{a_t}(\langle \mu_{\tilde{t}}, X_{t, a_t}\rangle) - \mathcal{F}_{a_t}(\langle \mu^*, X_{t, a_t}\rangle)}_{(VII)} \ .$$

Since $a_t$ is chosen greedily in Alg. 1 we have $(IV) \leq 0$. Then, applying Lemma 5.1(i), Cauchy-Schwarz and $\|X_{t,a}\|_* \leq \|x_{\max}\|_*$ for (I) and (VII) and Lemma 5.1(ii) for (II) and (VI), we obtain

$$(I) + (VII) \leq 2M\|\mu^* - \mu_{\tilde{t}}\|\|x_{\max}\|_* \ , \quad (II) + (VI) \leq 4M\|\mu^* - \mu_{\tilde{t}}\|\|x_{\max}\|_* \ .$$

By noticing that $\hat{\mathcal{F}}_{t, a}(\cdot)$ is the empirical CDF of the random variable $\langle \mu_{\tilde{t}}, X_a \rangle$ conditioned to $\mu_{\tilde{t}}$, we can bound (III) and (V) directly using the DKWM inequality (see Lemma C.1), which gives that with probability at least $1 - \delta/4$ and for all $t$ such that $3 \leq t \leq T$ we have

$$(III) + (V) \leq 2\sqrt{\frac{\log(8KT/\delta)}{t - 1}} \ .$$

We conclude the proof by combining the previous bounds and setting $\|\cdot\| = \|\cdot\|_{V_{\tilde{t}}}$ . $\qquad\square$

We proceed by controlling the term $\|\mu^* - \mu_{\tilde{t}}\|_{V_{\tilde{t}}} \|x_{\max}\|_{V_{\tilde{t}}^{-1}}$ in Lemma 5.2. The quantity $\|\mu^* - \mu_{\tilde{t}}\|_{V_{\tilde{t}}}$ can be bounded using the OFUL confidence bounds [1, Theorem 2], since the noise term in $\mu_{\tilde{t}}$ decreases at an appropriate rate. Controlling $\|x_{\max}\|_{V_{\tilde{t}}^{-1}}$ requires instead different results than the ones in [1], since it depends on the distributions of $\{X_a\}_{a=1}^K$ and not only on previous contexts and rewards. Hence, to provide an upper bound for $\|x_{\max}\|_{V_{\tilde{t}}^{-1}}$ which decreases with $t$, we also rely on Assumption A(iii) and the structure of Alg. 1, which enable the following history-agnostic lower bound on the probability of selecting one arm.

**Proposition 5.1.** *Let Assumption A hold, $a_t$ be generated by Alg. 1 and $c \in [0,1)$. Then with probability at least $1 - \delta/4$, for all $a \in [K]$ and all $t \geq 3 + 8 \log^{3/2}\left(5K\,e/\delta\right)/\left(1 - \sqrt[K]{c}\right)^3$ we have*

$$\mathbb{P}(a_t = a \,|\, \mathcal{H}_{t-1}^-) \geq \frac{c}{K} \quad,$$

*where we recall that $\mathcal{H}_t^- = \cup_{i=1}^t \left\{ \{X_{i,a}\}_{a=1}^K, r_{i,a_i}, a_i \right\}$.*

*Proof sketch (proof in Appendix C.2).* For any $a \in [K]$, let $\hat{r}_{t,a} = \langle \mu_{\tilde{t}}, X_{t,a} \rangle$ be the estimated reward for arm $a$ at round $t$, denote with $\mathcal{F}_{\hat{r}_{t,a}}$ the CDF of $\hat{r}_{t,a}$ conditioned on $\mu_{\tilde{t}}$, and let

$$\phi_{t,a} := \mathcal{F}_{\hat{r}_{t,a}}(\hat{r}_{t,a}) \quad, \quad \text{and} \quad \hat{\phi}_{t,a} := \hat{\mathcal{F}}_{t,a}(\hat{r}_{t,a}) \quad,$$

where $\hat{\mathcal{F}}_{t,a}(\hat{r}_{t,a})$ is defined in line 6 of Alg. 1. Now, by the definition of $a_t$ (line 7 of Alg. 1), we have

$$\mathbb{P}(a_t = a \,|\, \mathcal{H}_{t-1}^-) = \sum_{m=1}^K \frac{1}{m} \mathbb{P}(a \in C_t, |C_t| = m \,|\, \mathcal{H}_{t-1}^-) \quad,$$

where we introduced $C_t := \operatorname{argmax}_{a \in [K]} \hat{\phi}_{t,a}$. Let $\epsilon_t > 0$ and continue the analysis conditioning on the events where $\sup_{a \in [K]} |\phi_{t,a} - \hat{\phi}_{t,a}| \leq \epsilon_t$. Then, we can write

$$\mathbb{P}(a_t = a \,|\, \mathcal{H}_{t-1}^-) \geq \mathbb{P}(\hat{\phi}_{t,a} > \hat{\phi}_{t,a'}, \ \forall a' \neq a \,|\, \mathcal{H}_{t-1}^-) \geq \mathbb{P}(\phi_{t,a} > \phi_{t,a'} + 2\epsilon_t, \ \forall a' \neq a \,|\, \mathcal{H}_{t-1}^-) \quad,$$

where in the first inequality we considered the case when $a \in C_t$ and $|C_t| = 1$, and in the second inequality we considered the worst case scenario where $\hat{\phi}_{t,a} = \phi_{t,a} - \epsilon_t$ and $\hat{\phi}_{t,a'} = \phi_{t,a'} + \epsilon_t$. Assumption A(iv) and the additive noise in $\mu_{\tilde{t}}$ imply that $\langle \mu_{\tilde{t}}, X_a \rangle$ is an absolutely continuous random variable for each $a \in [K]$, which yields that $\{\phi_{t,a}\}_{a \in [K]}$ is uniformly distributed on $[0,1]$. Furthermore, $\{\phi_{t,a}\}_{a \in [K]}$ are also independent due to Assumption A(iii). Thus we have

$$\mathbb{P}(a_t = a \,|\, \mathcal{H}_{t-1}^-) \geq \int_0^1 \left(\mathbb{P}(\phi_{t,a'} < \mu - 2\epsilon_t)\right)^{K-1} \mathrm{d}\mu = \int_{2\epsilon_t}^1 (\mu - 2\epsilon_t)^{K-1} \mathrm{d}\mu = \frac{(1 - 2\epsilon_t)^K}{K} \quad.$$

Finally, thanks to Assumption A(ii) we can invoke the DKWM inequality to appropriately bound $\epsilon_t$ in high probability for all $t$ sufficiently large. $\qquad \square$

The property in Proposition 5.1 guarantees that, for sufficiently large $t$, the policy can get arbitrarily close to satisfy history-agnostic demographic parity in (3). In particular this allows us to control $\|x_{\max}\|_{V_{\tilde{t}}^{-1}}$ by using a standard matrix concentration inequality [33, Theorem 3.1] on a special decomposition of $V_{\tilde{t}}$, thereby enabling the following result (proof in Appendix C.3).

**Lemma 5.3.** *Let Assumption A hold, $a_t$ be generated by Alg. 1, $\tau_1 = 32K^3 \log^{3/2}\left(5K\,e/\delta\right)$, $\tau_2 = \frac{54L^2}{\lambda_{\min}^+(\Sigma)} \log(\frac{4d}{\delta})$ and $\tau = 4 \max(\tau_1, \tau_2) + 3$. Then, with probability at least $1 - \frac{3\delta}{4}$, for all $t \geq \tau$ we have*

$$\|\mu^* - \mu_{\tilde{t}}\|_{V_{\tilde{t}}} \|x\|_{V_{\tilde{t}}^{-1}} \leq \frac{8L}{\sqrt{\lambda_{\min}^+(\Sigma) \cdot t}} \left[ b_1 \sqrt{d \log((8 + 4t \max(L^2/\lambda, 1))/\delta)} + \lambda^{\frac{1}{2}} \|\mu^*\|_2 \right] \quad,$$

*where $b_1 = \lambda^{\frac{1}{2}} + R + L$, $\Sigma := K^{-1} \sum_{a=1}^K \mathbb{E}\left[X_a X_a^\top\right]$ and $\lambda_{\min}^+(\Sigma)$ is its smallest nonzero eigenvalue.*

Finally we obtain the desired high probability regret bound by combining Lemma 5.2 with Lemma 5.3 and summing over the $T$ rounds (see Appendix C.4 for a proof).

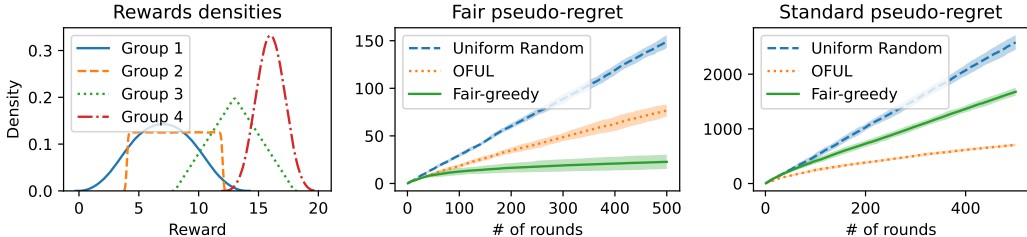

Figure 1: **Simulation Results**. First image is a density plot of the reward distributions while the second and third plot show the standard and fair pseudo-regrets, with mean (solid lines) $\pm$ standard deviation (shaded region) over 10 runs. To approximate the true reward CDF for each group we use the empirical CDF with $10^7$ samples.

**Theorem 1.** *Let Assumption A hold and $a_t$ be generated by Alg. 1. Then, with probability at least $1 - \delta$, for any $T \geq 1$ we have*

$$R_F(T) \leq \frac{96ML}{\sqrt{\lambda_{\min}^+(\Sigma)}} \left[ (\lambda^{\frac{1}{2}} + R + L)\sqrt{dT \log((8 + 4T \max(L^2/\lambda, 1))/\delta)} + \sqrt{\lambda T}\|\mu^*\|_2 \right]$$

$$+ 8\sqrt{\frac{T \log(8KT/\delta)}{3}} + \tau \ ,$$

*with $\tau$ defined in Lemma 5.3. Hence $R_F(T) = O(K^3 \log^{3/2}(K/\delta) + \sqrt{dT \log(KT/\delta)})$.*

The regret bound in Thm. 1 has two terms. The $O(K^3 \log^{3/2}(K))$ term describes the rounds needed to satisfy Proposition 5.1 with $c = 1/2$. The remaining part, which is of order $O(\sqrt{dT \log(KT)})$ is instead associated to the convergence of the empirical CDF and to the bandit performance. Indeed, it recalls the standard regret bound holding for finite-action linear contextual bandits [2, 12, 26].

## 6  Simulation with Diverse Reward Distributions

We present an illustrative proof of concept experiment which simulates groups with diverse reward distributions. We set $K = 4$, $\eta_t = 2\xi_t$, where $\xi_t$ has standard normal distribution, $X_a = B_a Y_a + c_a$ where each coordinate of $Y_a \in \mathbb{R}^4$ is an independent sample from the uniform distribution on $[0, 1]$, $B_a \in R^{(4K+1)\times 4}$ is such that $X_a$ contains $Y_a$ starting from the $4a$-th coordinate and $c_a$ has all the coordinates set to zero except for the last which is set to $3a$ to simulate a group bias. In this setup $\mu^*$ acts differently on each group, in particular, we note that $\mu^* \in R^{4K+1}$ has its last coordinate multiplying the group bias in $c_a$, which we set to 1, and 4 group-specific coordinates, which we set to manually picked values between 0 and 9. Results are shown in Fig. 1, where we compare our greedy policy in Alg. 1 with OFUL [1], both with regularization parameter set to 0.1, and with the Uniform Random policy. We observe that, as expected from our analysis, our policy achieves sublinear fair pseudo-regret, while also having better-than random, although linear, standard regret. Additional details and an experiment on US census data with gender as the sensitive group are in Appendix D.

## 7  Multiple Candidates for Each Group

In this section, we analyze the more realistic case where contexts from a given arm do not necessarily belong to the same group. The complete analysis is presented in Appendix E. In particular, we assume that at each round $t$, the agent receives $\{(X_{t,a}, s_{t,a})\}_{a=1}^K$, which are $K$ i.i.d. random variables where $s_{t,a} \in [G]$ is the sensitive group of the context $X_{t,a} \in \mathbb{R}^d$ and $G$ is the total number of groups. This setting can model for example a hiring scenario where at each round the employer has to choose among candidates belonging to different ethnic groups, some of which are minorities and hence have a small probability $\mathbb{P}(s_{t,a} = i)$ of being in the pool of received candidates. By naturally adapting the definition of fair-regret $R_F(T)$, the Fair-Greedy policy and Assumption A to this setting, with probability $1 - \delta$ we obtain the following regret bound (see Cor. E.1 in Appendix E).

$$R_{\mathrm{F}}(T) = O\left( \frac{G \log(GT/\delta)}{K q_{\min}} + \frac{(KG)^{3/2} \log^{3/2}(G/\delta)}{q_{\min}^{3/2}} + \sqrt{\frac{dT \log(GT/\delta)}{(1 + K/G) q_{\min}}} \right) \tag{4}$$

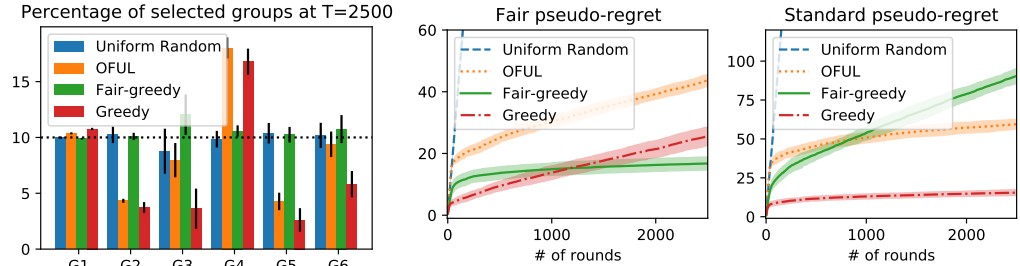

Figure 2: **US Census Results. Group = Ethnicity**. First image shows mean (colored bars) and std (thinner black bars), while the other two show the mean (solid lines) $\pm$ standard deviation (shaded region) over 10 runs. To compute the reward CDF for each group we use the empirical CDF on $5K$ samples from $D2$. Percentage of selected groups is computed by dividing the number of candidates of a given group selected by the policy by the total number of candidates of that group received by the agent. G$X$ with $X \in \{1, \ldots, 6\}$, stands for group $X$.

where $q_{\min} = \min_{i \in [G]} \mathbb{P}(s_{t,a} = i)G$, so that $q_{\min} = 1$ if and only if each group has equal probability of being sampled and $q_{\min} > 0$ without loss of generality. (4) is similar to Thm. 1, having the same dependency on $\delta$ and $T$ but an improved dependency on the number of arms $K$ when $K > G$, since contexts from all arms can be used to estimate the CDF of each group. The first term in (4) comes from the application of the Chernoff bound to lower bound the number of candidates in each group received by the agent, which is now random.

**US Census experiments. Group = Ethnicity.** We test this setting in practice by simulating the hiring scenario discussed above with data from the US Census containing the income and other useful indicators of several individuals in the United States. This data is accessed via the FolkTables library [13]. In particular, at each round, we sample $K = 10$ candidates at random from the population containing the $G = 6$ largest ethnic groups[3], the reward is a previously computed linear estimate of the income, while the noisy reward is the true reward plus some small gaussian noise. We compare the Fair-Greedy Policy with OFUL [1], Greedy (selects the candidate with the best estimated reward) and Uniform Random in Fig. 2. Similarly to the synthetic experiment in Sec. 6, the Fair-greedy policy achieves the best fair pseudo-regret and standard regret better than Uniform Random. Note that Greedy outperforms OFUL, which is too conservative in this scenario. Furthermore, the Fair-Greedy policy selects approximately the same percentage of candidates from each group, similarly to Uniform Random, while OFUL and Greedy select smaller percentages from G2, G3, G5 and G6. In Appendix E.1 we provide more details and a comparison with two oracle fair policies which shows that knowing $\mu^*$ plays a more important role than knowing the true reward CDFs of each group.

# 8 Conclusions and Future Work

We introduced the concept of group meritocratic fairness in linear contextual bandits, which states that a fair policy should select, at each round, the candidate with the highest relative rank in the pool. This allows us to compare candidates coming from different sensitive groups, but it is hard to satisfy since the relative rank is not directly observed and depends on both the underlying reward model and on the rewards distribution for each group. After defining an appropriate fair pseudo-regret we analyzed a greedy policy and proved that its fair pseudo-regret is sublinear with high probability.

This result was possible since we restricted the analysis to the case where the contexts of different groups are independent random variables and the rewards are absolutely continuous. Relaxing these assumptions is a challenging avenue for future work. In particular, without the independence of contexts across arms, different approaches relying on confidence intervals might be necessary. Other two interesting directions are (i) to study the optimality of the proposed results and establishing lower bounds for any algorithm which minimises the fair pseudo-regret and (ii) to design a learning policy which aims at achieving a tradeoff between group meritocratic fairness and reward maximization.

**Acknowledgments.** This work was supported in part by the EU Projects ELISE and ELSA.

---

[3]We remove groups with less than 5K individuals to compute accurately the true CDFs for the fair regret.

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
