# Appendices

Below we give an overview of the structure of the Appendices.

- In Appendix A we present some auxiliary lemmas which are used to prove our results.
- In Appendix B we present the proof of Lemma 4.1.
- Appendix C presents the proofs for the results in Sec. 5.
- In Appendix D we include additional details on the experimental setting in Sec. 6 and an experiment on the US Census Data with gender as sensitive attribute.
- In Appendix E we provide a more rigorous treatment of the setting introduced in Sec. 7, where we can have multiple condidates for each group at any given round. We also provide more details and additional comparisons for the experiment on the US Census Data in Sec. 7.
- In Appendix F we investigate the tradeoff between fair an standard regret.

## A    Auxiliary Lemmas

**Lemma A.1.** *Let $n \in \mathbb{N}$, and assume that $Y, \nu$ are independent random variables in $\mathbb{R}^n$, such that $Y$ is absolutely continuous and $\nu \neq 0$, almost surely. Then, $\nu^\top Y$ is an absolutely continuous random variable.*

*Proof.* It is enough to show that for any $A \subset \mathbb{R}$ with zero Lebesgue measure, $\mathbb{P}(\nu^\top Y \in A) = 0$. Let $A \subseteq \mathbb{R}$, then we can write

$$\mathbb{P}(\nu^\top Y \in A) = \mathbb{E}[\mathbb{P}(\nu^\top Y \in A \,|\, \nu)] \ .$$

We proceed the proof by controlling the term $\mathbb{P}(\nu^\top Y \in A \,|\, \nu)$. We know that $\nu \neq 0$ almost surely. Now, since $Y$ and $\nu$ are independent, let $\nu = w$ for a fixed $w \in \mathbb{R}^n$ such that $w \neq 0$, then we have that

$$\mathbb{P}(w^\top Y \in A) = \int_{y \in \mathbb{R}^n} \mathbb{1} \left\{ \frac{w^\top}{\|w\|_2} Y \in A' \right\} f_Y(y) \, \mathrm{d}y \ ,$$

where we defined $A' := \left\{ \frac{x}{\|w\|_2} : x \in A \right\}$. Now consider the change of basis matrix $R = (v_1, \ldots, v_n)^\top$, such that $v_1 = \frac{w}{\|w\|_2}$, with $RR^\top = \mathbb{I}_n$. By assigning $\hat{Y} = R^\top Y$, we can write

$$\mathbb{P}(w^\top Y \in A) = \int_{\hat{y} \in \mathbb{R}^n} \mathbb{1} \left\{ \hat{y}_1 \in A' \right\} f_Y(R\hat{y}) \, \mathrm{d}\hat{y} \ .$$

Since we assume that $Y$ is an absolutely continuous random variable, there exists $M_Y > 0$, such that $\sup_{y \in \mathbb{R}^n} f_Y(y) \leq M_Y$ almost surely, which allows us to write

$$\mathbb{P}(w^\top Y \in A) \leq M_Y \int_{\hat{y} \in \mathbb{R}^n} \mathbb{1} \left\{ \hat{y}_1 \in A' \right\} \, \mathrm{d}\hat{y} \ .$$

Finally, it is straightforward to check that if $A$ has a zero Lebesgue measure, then $A'$ also has a a zero Lebesgue measure, which gives $\mathbb{P}(w^\top Y \in A) = 0$. $\qquad\square$

**Lemma A.2** (Lipschitz CDF). *Let $n \in \mathbb{N}$, $\nu \in \mathbb{R}^n/\{0\}$ and $b \in \mathbb{R}$. Let also $Y$ be an absolutely continuous random variable with values in $\mathbb{R}^n$, with probability density function $f_Y$. Then the CDF of $Z = \langle \nu, Y \rangle + b$, namely $\mathcal{F}_Z$, is Lipschitz continuous. More specifically*

$$|\mathcal{F}_Z(r) - \mathcal{F}_Z(r')| \leq M'|r - r'| \qquad \forall r, r' \in \mathbb{R} \ ,$$

*where $M' = \max_{z \in \mathbb{R}} f_Z(z)$.*

*Proof.* Since $\nu \neq 0$ and $Y$ is absolutely continuous, $Z$ is also absolutely continuous with probability density $f_Z$ (see Lemma A.1). Furthermore, if $r' \leq r$, we can write

$$\mathcal{F}_Z(r) - \mathcal{F}_Z(r') = \int_{-\infty}^{r} f_Z(t)\,\mathrm{d}t - \int_{-\infty}^{r'} f_Z(t)\,\mathrm{d}t = \int_{r'}^{r} f_Z(t)\,\mathrm{d}t \leq M'(r - r').$$

Applying the same reasoning to the case when $r \leq r'$ concludes the proof. $\qquad\square$

**Lemma A.3.** *Let $\{X_a\}_{a=1}^{K}$ be $K$ random variables with values in $\mathbb{R}^d$ and such that they are all $0$ with probability strictly less than one. Define $\Sigma = K^{-1}\sum_{a=1}^{K}\mathbb{E}[X_a X_a^\top]$ and let $\Sigma = USU^\top$ be its compact eigenvalue decomposition with $U \in \mathbb{R}^{d\times r}$, $S \in \mathbb{R}^{r\times r}$ with $1 \leq r \leq d$. Assume that $S$ is invertible. Then, for any $y \in \cup_{a=1}^{K}\mathrm{Supp}(X_a)$, we have $UU^\top y = y$ and $\lambda_{\min}^+(\Sigma)\|y\|_2^2 \leq y^\top\Sigma y$, where $\lambda_{\min}^+(\Sigma)$ is the smallest non-zero eigenvalue of the matrix $\Sigma$.*

*Proof.* Let $X$ be a random variable with the distribution $\mathbb{P}(X) = K^{-1}\sum_{a=1}^{K}\mathbb{P}(X_a)$. It is straightforward to check that $\Sigma = \mathbb{E}[XX^\top]$, and $y \in \mathrm{Supp}(X)$. We can also write $y = y_1 + y_2$ where $y_2 \in \mathrm{Ker}(\Sigma) := \{z \in \mathbb{R}^d : \Sigma z = 0\}$ and $y_1 \in \mathrm{Ker}(\Sigma)^\perp := \{z \in \mathbb{R}^d : \langle z, x\rangle = 0, \forall x \in \mathrm{Ker}(\Sigma)\}$. This implies that

$$y_2^\top\Sigma y_2 = \mathbb{E}[y_2^\top XX^\top y_2] = 0 \ . \tag{5}$$

Now, let $f(x) = (y_2^\top x)^2$. Then, $f(x) \geq 0$, for any $x \in \mathbb{R}^d$ and $f(y) = \|y_2\|_2^4$. Furthermore, since, $f(x)$ is a continuous function there exists $\epsilon > 0$, such that for any $z \in B(y, \epsilon) = \{x \in \mathbb{R}^d : \|x - y\|_2 < \epsilon\}$, $f(z) \geq \frac{\|y_2\|_2^4}{2}$. On the other hand, since $y \in \mathrm{Supp}(X)$, $\mathbb{P}(X \in B(y, \epsilon)) > 0$. Hence, we can write

$$0 = y_2^\top\Sigma y_2 = \mathbb{E}[f(X)] \geq \mathbb{E}[f(X)\mathbb{1}\{X \in B(y,\epsilon)\}] \geq \frac{\|y_2\|_2^4}{2}\mathbb{P}(X \in B(y,\epsilon)) \ ,$$

therefore $y_2 = 0$ which implies that $y \in \mathrm{Ker}(\Sigma)^\perp$. Since $UU^\top y$ is the orthogonal projection of $y$ onto $\mathrm{Ker}(\Sigma)^\perp$ we conclude that $y = UU^\top y$, $y^\top = y^\top UU^\top$ and

$$y^\top\Sigma y = y^\top USU^\top y \geq \lambda_{\min}^+(\Sigma)y^\top UU^\top y = \lambda_{\min}^+(\Sigma)\|y\|_2^2 \ .$$

$\qquad\square$

## B  Proof of Lemma 4.1

*Proof.* If $t < 3$ then $\mu_{\tilde{t}} = 0$ and $a_t \sim \mathcal{U}[[K]]$ and the statement follows. If $t \geq 3$, Let $\mu \in \mathcal{S} := \{\mu' \in \mathbb{R}^d : \mu'^\top B_a \neq 0 \,\forall a \in [K]\}$, $\hat{r}_{i,a} = \langle \mu, X_{i,a}\rangle$ and $t' = t - \tilde{t} - 1$. Then by Lemma A.1 $\hat{r}_{i,a}$ is absolutely continuous. Given a permutation of indices $j = (j_1, \ldots, j_{t'})$ where $j_i \in \{\tilde{t} + 1, \ldots, t\}$, for $i \in [t']$. Let $\Omega_a$ be the set of the events of $\{X_{i,a}\}_{i=\tilde{t}+1}^{t}$ and $P$ be the set of all permutations of the indices $\{\tilde{t} + 1, \ldots, t\}$. Consider the event

$$E_{a,j} = \{\omega \in \Omega_a : \hat{r}_{j_1,a} < \cdots < \hat{r}_{j_{t'},a}\} \ .$$

Since $\{\hat{r}_{i,a}\}_{i=\tilde{t}+1}^{t}$ are absolutely continuous, we have for all $k \neq i$, $\mathbb{P}(\hat{r}_{j_i,a} = \hat{r}_{j_k,a}) = 0$ and this yields $\Omega_a = \cup_{j\in P}E_{a,j}$ and $E_{a,j} \cap E_{a,j'} = \emptyset$ for all $j \neq j'$. Furthermore, since $\{\hat{r}_{i,a}\}_{i=\tilde{t}+1}^{t}$ are i.i.d. we have that $p_a := \mathbb{P}(E_{a,j}) = \mathbb{P}(E_{a,j'})$ for all $j \neq j'$. In particular, since $|P| = t'!$ we have $p_a = 1/(t'!)$.

Let $\phi_a = (t' - 1)^{-1}\sum_{i=\tilde{t}+1}^{t-1}\mathbb{1}\{\hat{r}_{i,a} < \hat{r}_{t,a}\}$. Let $b \in \{0, \ldots, t-1\}$ and let $P_b = \{j \in P : j_{b+1} = t\}$. We have that $|P_b| = (t' - 1)!$ and

$$\mathbb{P}(\phi_a = b/(t' - 1)) = \sum_{j\in P_b}\mathbb{P}(E_{a,j}) = (t' - 1)!p_a = \frac{1}{t'} \ .$$

As a consequence, for all $a \in [K]$, $\phi_a$ is uniform over $\{0, 1/(t'-1), \ldots, 1\}$. Since $\{r_{i,a}\}_{i\in[t+1], a\in[K]}$ are mutually independent we have that $\{\phi_a\}_{a\in[K]}$ are i.i.d. discrete uniform random variables. As a

consequence, let $\hat{a} = \mathcal{U}\left[\text{argmax}_{a' \in [K]} \hat{\phi}_a\right]$ we have that $\mathbb{P}(\hat{a} = a) = 1/K$. Using the definition of $\hat{a}$ we have

$$\mathbb{P}(a_t = a) = \frac{1}{K}\mathbb{P}(\mu_{\tilde{t}} \in \mathcal{S}) + \mathbb{P}(a_t = a \mid \mu_{\tilde{t}} \in \mathcal{S}^c)\mathbb{P}(\mu_{\tilde{t}} \in \mathcal{S}^c) = \frac{1}{K} \; ,$$

where the last equality is derived by the fact that by the construction of $\mu_{\tilde{t}}$, $\mathbb{P}(\mu_{\tilde{t}} \in \mathcal{S}) = 1$. □

## C  Proofs of Results in Sec. 5

The following result is used in the Proof of Lemma 5.2. Its proof is obtained by using the Dvoretzky–Kiefer–Wolfowitz-Massart inequality [15, 30] combined with a union bound.

**Lemma C.1.** *Let Assumption A(ii) hold and $\hat{\mathcal{F}}_{t,a}(r)$, and $\mu_{\tilde{t}}$ to be generated by Alg. 1. Let $Z_a := \langle \mu_{\tilde{t}}, X_a \rangle$ and denote with $\mathcal{F}_{Z_a}$ its CDF, conditioned on $\mu_{\tilde{t}}$. Then, with probability at least $1 - \delta$ we have that for all $3 \le t \le T$*

$$\sup_{a \in [K], r \in \mathbb{R}} |\hat{\mathcal{F}}_{t,a}(r) - \mathcal{F}_{Z_a}(r)| \le \sqrt{\frac{\log(2KT/\delta)}{t-1}} \; .$$

*Proof.* Let $3 \le t \in T$ and recall that $\tilde{t} = \lfloor (t-1)/2 \rfloor$. Note that from Assumption A(ii), for all $a \in [K]$, $\{\langle \mu_{\tilde{t}}, X_{i,a} \rangle\}_{i=\tilde{t}+1}^{t-1}$ are i.i.d copies of $Z_a$, conditioned on $\mu_{\tilde{t}}$. Since $\hat{\mathcal{F}}_{t,a}(r)$ is the empirical CDF of $Z_a$ conditioned on $\mu_{\tilde{t}}$, we can apply the Dvoretzky–Kiefer–Wolfowitz-Massart inequality [15, 30] to obtain

$$\mathbb{P}\left(\sup_{r \in \mathbb{R}} |\hat{\mathcal{F}}_{t,a}(r) - \mathcal{F}_{Z_a}(r)| \ge \sqrt{\frac{\log(2/\delta')}{2(t-1-\tilde{t})}}\right) \le \delta' \; .$$

Therefore, since $\tilde{t} \le (t-1)/2$ we deduce that $\mathbb{P}[E_{t,a}] \le \delta'$ , where

$$E_{t,a} = \left\{ \{X_{i,a}\}_{i=\tilde{t}+1}^{t-1}, \mu_{\tilde{t}} : \sup_{r \in \mathbb{R}} |\hat{\mathcal{F}}_{t,a}(r) - \mathcal{F}_{Z_a}(r)| \ge \sqrt{\frac{\log(2/\delta)}{t-1}} \right\} \; .$$

Consequently, by applying a union bound we obtain

$$\mathbb{P}\left[\cup_{i=1}^T \cup_{a=1}^K E_{t,a}\right] \le \sum_{i=1}^T \sum_{a=1}^K \mathbb{P}(E_{t,a}) \le KT\delta' \; ,$$

Finally, by substituting $\delta' = \delta/(KT)$ and computing the probability of the complement of $\cup_{i=1}^T \cup_{a=1}^K E_{t,a}$, we obtain the desired result. □

### C.1  Proof of Lemma 5.1

*Proof.* For every $a \in [K]$, by Assumption A(iv), we have that $X_a = B_a Y_a + c_a$ where $Y_a \in \mathbb{R}^{d_a}$ is absolutely continuous with density $f_a$. Let $\nu^* := \mu^{*\top} B_a$, $\nu := \mu^\top B_a$, $b^* := \langle \mu^*, c_a \rangle$, $b := \langle \mu, c_a \rangle$. Then we have

$$Z_a = \langle \mu^*, X_a \rangle = \langle \nu^*, Y_a \rangle + b^*, \quad \text{and} \quad \tilde{Z}_a = \langle \mu, X_a \rangle = \langle \nu, Y_a \rangle + b \; .$$

From Assumption A(iv) we also have that $\nu^* \ne 0$, hence, by applying Lemma A.2 with $\nu = \nu^*$ and $Y = Y_a$ and by taking the maximum over $a \in [K]$, the statement (i) follows.

We now prove (ii). Since $Y_a$ is absolutely continuous we can write for any $r \in \mathbb{R}$

$$|\mathcal{F}_a(r) - \mathcal{F}_{\tilde{Z}_a}(r)| = |\mathcal{F}_{\langle \nu^*, Y_a \rangle + b^*}(r) - \mathcal{F}_{\langle \nu, Y_a \rangle + b}(r)|$$

$$\le \int_{y \in \mathbb{R}^{d_a}} \left| \mathbb{1}\{\langle \nu^*, y \rangle + b^* \le r\} - \mathbb{1}\{\langle \nu, y \rangle + b \le r\} \right| f_a(y) \, \mathrm{d}y \; .$$

Now, by adding and subtracting $q(y) := \langle \nu^* - \nu, y \rangle + b^* - b$ and letting $r' := r - b^*$ we have

$$|\mathcal{F}_a(r) - \mathcal{F}_{\tilde{Z}_a}(r)| \leq \int_{y \in \mathbb{R}^{d_a}} \left| \mathbb{1}\left\{\langle \nu^*, y \rangle \leq r'\right\} - \mathbb{1}\left\{\langle \nu^*, y \rangle \leq r' + q(y)\right\} \right| f_a(y) \, \mathrm{d}y$$

$$\leq \int_{y \in \mathbb{R}^{d_a}} \mathbb{1}\left\{r' - |q(y)| \leq \langle \nu^*, y \rangle \leq r' + |q(y)|\right\} f_a(y) \, \mathrm{d}y \ .$$

By Cauchy-Schwarz inequality, for any $y \in \mathrm{Supp}(Y_a)$, we get

$$|q(y)| \leq |\langle \nu^* - \nu, y \rangle + b^* - b| \leq |\langle \mu^* - \mu, B_a y + c_a \rangle| \leq \|\mu^* - \mu\| \|x_{\max}\|_* \ ,$$

where we defined $\|x_{\max}\|_* := \max_{x \in \cup_{a=1}^K \mathrm{Supp}(X_a)} \|x\|_* = \max_{y \in \cup_{a=1}^K \mathrm{Supp}(Y_a)} \|B_a y + c_a\|_*$. We now let $\kappa = \|\mu^* - \mu\| \|x_{\max}\|_*$, and note that

$$|\mathcal{F}_a(r) - \mathcal{F}_{\tilde{Z}_a}(r)| \leq \int_{y \in \mathbb{R}^{d_a}} \mathbb{1}\left\{r' - \kappa \leq \langle \nu^*, y \rangle \leq r' + \kappa\right\} f_a(y) \, \mathrm{d}y \ .$$

To control the above integral, we provide a proper change of variables. To this end, since by the assumption $\nu^* \neq 0$, we let $\{v_1, \ldots, v_{d_a}\}$ be an orthonormal basis of $\mathbb{R}^{d_a}$, with $v_1 = \nu^*/\|\nu^*\|_2$. Moreover, let $R = (v_1, \ldots, v_{d_a})$ to be the corresponding change of basis matrix. Then, for all $y \in \mathbb{R}^{d_a}$, we can always write $y = R\hat{y}$, where $\hat{y}_i = \langle y, v_i \rangle$, with $\hat{y}_1 = \frac{\langle \nu^*, y \rangle}{\|\nu^*\|_2}$. Hence we denote with $\hat{Y}_a = R^\top Y_a$ which now has the first coordinate parallel to $\nu^*$. Using the change of variables formula for multivariate integrals and noting that we are applying a rotation and hence $|\det(R)| = 1$ and $f_{R\hat{Y}_a}(R\hat{y}) = f_{\hat{Y}_a}(\hat{y})/|\det(R)| = f_{\hat{Y}_a}(\hat{y})$ we get

$$|\mathcal{F}_a(r) - \mathcal{F}_{\tilde{Z}_a}(r)| \leq \int_{\hat{y} \in \mathbb{R}^{d_a}} \mathbb{1}\left\{\frac{r' - \kappa}{\|\nu^*\|_2} \leq \hat{y}_1 \leq \frac{r' + \kappa}{\|\nu^*\|_2}\right\} f_{\hat{Y}_a}(\hat{y}) \, \mathrm{d}\hat{y} \ .$$

Let $z = (\hat{y}_2, \ldots, \hat{y}_{d_a})$. By Fubini's Theorem, and with the convention that $f_{\hat{Y}_a}(\hat{y}_1, z) = f_{\hat{Y}}(\hat{y}_1, z_1, \ldots, z_{d_a - 1})$ we have

$$|\mathcal{F}_a(r) - \mathcal{F}_{\tilde{Z}_a}(r)| \leq \int_{\hat{y}_1 \in \mathbb{R}} \mathbb{1}\left\{\frac{r' - \kappa}{\|\nu^*\|_2} \leq \hat{y}_1 \leq \frac{r' + \kappa}{\|\nu^*\|_2}\right\} \int_{z \in \mathbb{R}^{d_a - 1}} f_{\hat{Y}_a}(\hat{y}_1, z) \, \mathrm{d}z \, \mathrm{d}\hat{y}_1$$

$$= \int_{\hat{y}_1 \in \mathbb{R}} \mathbb{1}\left\{\frac{r' - \kappa}{\|\nu^*\|_2} \leq \hat{y}_1 \leq \frac{r' + \kappa}{\|\nu^*\|_2}\right\} f_{\hat{Y}_1}(\hat{y}_1) \, \mathrm{d}\hat{y}_1 \ ,$$

where $f_{\hat{Y}_1}(\hat{y}_1) := \int_{z \in \mathbb{R}^{d_a - 1}} f_{\hat{Y}}(\hat{y}_1, z_1, \ldots, z_{d_a - 1}) \, \mathrm{d}z$ is the marginal density of $\hat{Y}_1 = \frac{\langle \nu^*, Y_a \rangle}{\|\nu^*\|_2}$, and we highlight that $\hat{Y}_1 = (Z_a - b^*)/\|\nu^*\|_2$. Finally note that

$$\max_{y_1 \in \mathbb{R}} f_{\hat{Y}_1}(y_1) = \|\nu^*\|_2 \max_{y_1 \in \mathbb{R}} f_{Z_a}(y_1) = \|\nu^*\|_2 M \ ,$$

which yields

$$|\mathcal{F}_a(r) - \mathcal{F}_{\tilde{Z}_a}(r)| \leq 2\kappa M \ ,$$

and (ii) follows by substituting the definition of $\kappa$. $\qquad\square$

## C.2  Proof of Proposition 5.1

*Proof.* Recall the definition of Alg. 1. For any $a \in [K]$ let $\hat{r}_{t,a} = \mu_{\tilde{t}}^\top X_{t,a}$, which is the estimated reward for arm $a$, at round $t$. Note that $\mu_{\tilde{t}}$ and $X_{t,a}$ are independent random variables. Furthermore, denote with $\mathcal{F}_{\hat{r}_{t,a}}$ the CDF of $\hat{r}_{t,a}$ conditioned on $\mu_{\tilde{t}}$, and let

$$\phi_{t,a} := \mathcal{F}_{\hat{r}_{t,a}}(\hat{r}_{t,a}) \ , \quad \text{and} \quad \hat{\phi}_{t,a} := \hat{\mathcal{F}}_{t,a}(\hat{r}_{t,a}) \ .$$

Now, by the definition of the algorithm, we have

$$\mathbb{P}(a_t = a \mid \mathcal{H}_{t-1}^-) = \sum_{m=1}^K \frac{1}{m} \mathbb{P}(a \in C_t, |C_t| = m \mid \mathcal{H}_{t-1}^-) \ ,$$

where we introduced $C_t := \text{argmax}_{a \in [K]} \hat{\phi}_{t,a}$. Let $\epsilon_t > 0$ and continue the analysis conditioning on the events where $\sup_{a \in [K]} |\phi_{t,a} - \hat{\phi}_{t,a}| \le \epsilon_t$. Then, we can write

$$\mathbb{P}(a_t = a \,|\, \mathcal{H}_{t-1}^-) \ge \mathbb{P}(\hat{\phi}_{t,a} > \hat{\phi}_{t,a'} \,,\, \forall a' \ne a \,|\, \mathcal{H}_{t-1}^-) \ge \mathbb{P}(\phi_{t,a} > \phi_{t,a'} + 2\epsilon_t \,,\, \forall\, a' \ne a \,|\mathcal{H}_{t-1}^-) \,,$$

where in the first inequality we considered the case when $a \in C_t$ and $|C_t| = 1$. In the second inequality we considered the worst case scenario where $\hat{\phi}_{t,a} = \phi_{t,a} - \epsilon_t$ and $\hat{\phi}_{t,a'} = \phi_{t,a'} + \epsilon_t$. Recall that by the construction of the algorithm $\mu_{\tilde{t}} = V_{\tilde{t}}^{-1} X_{1:\tilde{t}}^{\top} r_{1:\tilde{t}} + (1/\sqrt{d\tilde{t}}) \cdot \gamma_{\tilde{t}}$. for all $a \in [K]$, the additive noise $(1/\sqrt{d\tilde{t}})\gamma_{\tilde{t}}$ assures that $\mu_{\tilde{t}}^{\top} B_a \ne 0$, almost surely. Therefore, by Lemma A.1 $\hat{r}_{t,a} = \langle \mu_{\tilde{t}}, X_{t,a} \rangle$ conditioned on $\mu_{\tilde{t}}$ is absolutely continuous.

Assumption A(iii) and [9, Theorem 2.1.10] yield that $\{\phi_{t,a}\}_{a \in [K]}$ are independent and uniformly distributed on $[0, 1]$ and in turn that

$$\mathbb{P}(a_t = a \,|\, \mathcal{H}_{t-1}^-) \ge \int_0^1 (\mathbb{P}(\phi_{t,a'} < \mu - 2\epsilon_t))^{K-1} \, \mathrm{d}\mu = \int_{2\epsilon_t}^1 (\mu - 2\epsilon_t)^{K-1} \, \mathrm{d}\mu = \frac{(1 - 2\epsilon_t)^K}{K}. \quad (6)$$

We continue by computing an $\epsilon_t$ for which $\sup_{a \in [K]} |\phi_{t,a} - \hat{\phi}_{t,a}| \le \epsilon_t$ holds with high probability. Observing that, conditioned on $\mu_{\tilde{t}}$, $\hat{\mathcal{F}}_{t,a}$ is the empirical CDF of $\mathcal{F}_{\hat{r}_{t,a}}$, we can use the Dvoretzky–Kiefer–Wolfowitz-Massart inequality to obtain, for any $a \in [K]$, $t \ge 3$, and $s \ge 0$

$$\mathbb{P}\left(|\phi_{t,a} - \hat{\phi}_{t,a}| \ge s\right) \le 2\exp\left(-2s^2(t - \tilde{t} - 1)\right) \,.$$

Now, let $\tau_0 := 3 + 8 \log^{3/2} \left(5K \,\mathrm{e}/\delta\right) / \left(1 - \sqrt[K]{c}\right)^3$. By applying the union bound, we can write

$$\mathbb{P}\left(\sup_{t \ge \tau_0, a \in [K]} |\phi_{t,a} - \hat{\phi}_{t,a}| \ge s\right) \le K \sum_{t=\tau_0}^{\infty} \mathbb{P}\left(|\phi_{t,a} - \hat{\phi}_{t,a}| \ge s\right)$$

$$\le 2K \sum_{t=\tau_0}^{\infty} \exp\left(-2s^2(t - \tilde{t} - 1)\right).$$

Since $\tilde{t} = \lfloor \frac{t-1}{2} \rfloor$, it is straightforward to check that

$$\mathbb{P}\left(\sup_{t \ge \tau_0, a \in [K]} |\phi_{t,a} - \hat{\phi}_{t,a}| \ge s\right) \le 2K \int_{t=\tau_0 - 1}^{\infty} \exp\left(-s^2 t\right) \, \mathrm{d}t \le 2Ks^{-2} \exp\left(-s^2(\tau_0 - 1)\right) \,.$$

Now, for any $\delta \in (0, 1)$, by assigning $s = \sqrt{\frac{\log(4K(\tau_0 - 1)/\delta)}{\tau_0 - 1}}$, we get

$$\mathbb{P}\left(\sup_{t \ge \tau_0, a \in [K]} |\phi_{t,a} - \hat{\phi}_{t,a}| \ge \sqrt{\frac{\log(4K(\tau_0 - 1)/\delta)}{\tau_0 - 1}}\right) \le \frac{\delta}{2 \log(4K(\tau_0 - 1)/\delta)} \le \frac{\delta}{4} \,, \quad (7)$$

where from $\tau_0 \ge 3, \delta < 1 \implies 4K(\tau_0 - 1)/\delta \ge 8 \ge \mathrm{e}^2 \implies \log(4K(\tau_0 - 1)/\delta) \ge 2$ we obtain the last inequality. From (6), it follows that

$$\inf_{t \ge \tau_0, a \in [K]} \mathbb{P}\left(a_t = a | \mathcal{H}_{t-1}^-\right) \ge \frac{(1 - 2\sup_{t \ge \tau} \epsilon_t)^K}{K} \,.$$

Moreover, form (7), by letting $\epsilon_t = \sqrt{\frac{\log(4K(\tau_0 - 1)/\delta)}{\tau_0 - 1}}$, with probability at least $1 - \frac{\delta}{4}$, we have

$$\inf_{t \ge \tau, a \in [K]} \mathbb{P}\left(a_t = a | \mathcal{H}_{t-1}^-\right) \ge \frac{1}{K}\left(1 - 2\underbrace{\sqrt{\frac{\log(4K(\tau_0 - 1)/\delta)}{\tau_0 - 1}}}_{(I)}\right)^K \,. \quad (8)$$

For the term (I) in the above, using $\log(x) \le \log(5\,\mathrm{e}/4)x^{1/3}$ and $x \ge x^{2/3}$ for any $x \ge 1$ we deduce that

$$(I) = \frac{\log(4K/\delta) + \log(\tau_0 - 1)}{\tau_0 - 1} \le \frac{\log(4K/\delta) + \log(5\,\mathrm{e}/4)}{(\tau_0 - 1)^{2/3}} = \frac{\log(5K\,\mathrm{e}/\delta)}{(\tau_0 - 1)^{2/3}} \,.$$

Now, by substituting $\tau_0 = 3 + 8 \log^{3/2} \left(5K\,\mathrm{e}/\delta\right) / \left(1 - \sqrt[K]{c}\right)^3$, we get that $(I) \le \frac{1}{4}\left(1 - \sqrt[K]{c}\right)^2$ and conclude the proof by plugging this inequality in (8). $\qquad \square$

## C.3 Proof of Lemma 5.3

We start by establishing some required lemmas.

**Lemma C.2.** *Let $\Sigma$, $\tau_1$, $\tau_2$ be defined in Lemma 5.3, $\tau_3 = \max(\tau_1, \tau_2)$, $\Sigma = USU^\top$ be the compact eigenvalue decomposition of $\Sigma$, with $U \in \mathbb{R}^{d\times r}$, $S \in \mathbb{R}^{r\times r}$ is a diagonal matrix with non-zero diagonal elements, and $U^\top U = \mathbb{I}_r$. Denote $\hat{S}_{t_0} = \sum_{i=1}^{\tilde{t}} U^\top X_{i,a_i} X_{i,a_i}^\top U$, where for $i \in [\tilde{t}]$, $a_i$ is given by Alg. 1. Then with probability at least $1 - \frac{\delta}{2}$, for any $t \geq 2\tau_3 + 3$ we have*

$$\lambda_{\min}\left(\hat{S}_{t_0}\right) \geq \frac{(\tilde{t} - \tau_3)\lambda_{\min}^+(\Sigma)}{4} \ .$$

*Proof.* Let $\tilde{S}_{t_0} := \sum_{i=1}^{\tilde{t}} \mathbb{E}\left[U^\top X_{i,a_i} X_{i,a_i}^\top U | \mathcal{H}_{i-1}^-\right]$. First, note that for any $\tau_3 \leq i \leq \tilde{t}$, we can write

$$\mathbb{E}\left[U^\top X_{i,a_i} X_{i,a_i}^\top U | \mathcal{H}_{i-1}^-\right] = \sum_{a=1}^{K} \mathbb{E}\left[U^\top X_{i,a_i} X_{i,a_i}^\top U | \mathcal{H}_{i-1}^-, a_i = a\right] \mathbb{P}\left(a_i = a \,|\, \mathcal{H}_{i-1}^-\right)$$

$$= \sum_{a=1}^{K} \mathbb{E}\left[U^\top X_{i,a} X_{i,a}^\top U\right] \mathbb{P}\left(a_i = a \,|\, \mathcal{H}_{i-1}^-\right) \ ,$$

where the last equality holds based on the fact that $X_{i,a_i} X_{i,a_i}^\top$ conditioned on $a_i$, is independent from $\mathcal{H}_{i-1}^-$. Then, since $t \geq 3 + 64K^3 \log^{3/2}(5K\,\mathrm{e}/\delta)$, by utilizing Proposition 5.1 with $c = \frac{1}{2}$ and noting that $1/(1 - \sqrt[K]{1/2}) \leq 2K$ for all $K \geq 1$, with probability at least $1 - \frac{\delta}{4}$, we have $\mathbb{P}(a_i = a \,|\, \mathcal{H}_{i-1}^-) \geq \frac{1}{2K}$. Therefore, with probability at least $1 - \frac{\delta}{4}$, we obtain

$$\lambda_{\min}\left(\mathbb{E}\left[U^\top X_{i,a_i} X_{i,a_i}^\top U \,|\, \mathcal{H}_{i-1}^-\right]\right) \geq \frac{1}{2}\lambda_{\min}\left(K^{-1}\sum_{a=1}^{K} U^\top \mathbb{E}\left[X_a X_a^\top\right] U\right) = \frac{\lambda_{\min}^+(\Sigma)}{2} \ ,$$

and consequently, with probability at least $1 - \frac{\delta}{4}$ we have

$$\lambda_{\min}(\tilde{S}_{\tilde{t}}) \geq \sum_{i=1}^{\tilde{t}} \lambda_{\min}(U^\top \mathbb{E}[X_{i,a_i} X_{i,a_i}^\top \,|\, \mathcal{H}_{i-1}^-]U) \geq (\tilde{t} - \tau_3) \cdot \frac{\lambda_{\min}^+(\Sigma)}{2} \ , \tag{9}$$

where in the last two displays we used the concavity attribute of the function $\lambda_{\min}(\cdot)$. Note that $\{X_{i,a_i}\}_{i=1}^{\infty}$, is an adaptive sequence with respect to the filtration $\{\mathcal{H}_i^-\}_{i=0}^{\infty}$, with

$$\|U^\top X_{i,a_i} X_{i,a_i}^\top U\|_{\mathrm{op}} \leq \|X_{i,a_i}\|_2^2 \leq L^2 \ ,$$

for any $i \in [\tilde{t}]$. Let $\iota = \tilde{t} - \tau_3$. Now, by invoking [33, Theorem 3.1] (with $\delta = \frac{1}{2}$ and $\mu = \lambda_{\min}^+(\Sigma)/2$, where $\delta, \mu$ are constants that appear in the latter theorem), we have

$$\mathbb{P}\left(\lambda_{\min}\left(\hat{S}_{t_0}\right) \leq \frac{\iota\lambda_{\min}^+(\Sigma)}{4} \quad \text{and} \quad \lambda_{\min}\left(\tilde{S}_{t_0}\right) \geq \frac{\iota\lambda_{\min}^+(\Sigma)}{2}\right) \leq d \cdot \left(\frac{e^{-\frac{1}{2}}}{\frac{1}{2}^{\frac{1}{2}}}\right)^{\frac{\iota\lambda_{\min}^+(\Sigma)}{4L^2}} \leq q \ ,$$

where we introduced $q = d \cdot \exp(-\frac{\iota\lambda_{\min}^+(\Sigma)}{27L^2})$, and we used the inequality $\mathrm{e}^{-\frac{1}{2}} \cdot \frac{1}{2}^{-\frac{1}{2}} \leq \mathrm{e}^{-\frac{4}{27}}$. Note that since $\tilde{t} \geq \tau_3 = \frac{54L^2}{\lambda_{\min}^+(\Sigma)}\log(\frac{4d}{\delta})$, we have $q \leq \frac{\delta}{4}$. Let $p = \mathbb{P}[\lambda_{\min}(\tilde{S}_{t_0}) \geq \frac{\iota\lambda_{\min}^+(\Sigma)}{2}]$, then we can write

$$\mathbb{P}\left(\lambda_{\min}\left(\hat{S}_{t_0}\right) \leq \frac{\iota\lambda_{\min}^+(\Sigma)}{4} \,\middle|\, \lambda_{\min}\left(\tilde{S}_{t_0}\right) \geq \frac{\iota\lambda_{\min}^+(\Sigma)}{2}\right) \leq \frac{\delta}{4p} \ ,$$

and accordingly

$$\mathbb{P}\left(\lambda_{\min}\left(\hat{S}_{t_0}\right) \geq \frac{\iota\lambda_{\min}^+(\Sigma)}{4} \quad \text{and} \quad \lambda_{\min}\left(\tilde{S}_{t_0}\right) \geq \frac{\iota\lambda_{\min}^+(\Sigma)}{2}\right) \geq 1 - \frac{\delta}{2} \ ,$$

where we used $p \geq 1 - \frac{\delta}{4}$, which follows from (9). Substituting $\iota = \tilde{t} - \tau_3$ gives the final result. $\square$

**Lemma C.3.** *Let $x \in \cup_{a=1}^{K} \mathrm{Supp}(X_a)$ and $\tau_3$ be defined in Lemma C.2, then with probability at least $1 - \frac{\delta}{2}$, for all $t \geq 2\tau_3 + 3$ we have*

$$\|x\|_{V_{\tilde{t}}^{-1}} \leq \frac{2L}{\sqrt{\lambda_{\min}^{+}(\Sigma)(\tilde{t} - \tau_3)}} \quad .$$

*Proof.* Note that if $x = 0$ it is straightforward to check that the statement holds. So without loss of generality we assume that $x \in \mathfrak{S}$, where $\mathfrak{S} = \cup_{a=1}^{K} \mathrm{Supp}(X_{t,a}) - \{0\}$. Consider the compact singular value decomposition $\Sigma = USU^{\top}$ where $U \in \mathbb{R}^{d \times r}$, $S \in \mathbb{R}^{r \times r}$ is a diagonal matrix with non-zero diagonal elements (due to Assumption A(iv)) and $U^{\top}U = \mathbb{I}_r$. Denote $\hat{S}_{\tilde{t}} = U^{\top}\hat{\Sigma}_{\tilde{t}}U$. For any $x \in \mathfrak{S}$ we have from Lemma A.3 that $UU^{\top}x = x$, and $x^{\top}UU^{\top} = x^{\top}$. First, we claim that

$$U^{\top}(\hat{\Sigma}_{\tilde{t}} + \lambda\mathbb{I}_d)^{-1}U = (\hat{S}_{\tilde{t}} + \lambda\mathbb{I}_r)^{-1} \quad .$$

To prove the above claim, it is enough to show that

$$(\hat{S}_{\tilde{t}} + \lambda\mathbb{I}_r)U^{\top}(\hat{\Sigma}_{\tilde{t}} + \lambda\mathbb{I}_d)^{-1}U = U^{\top}(\hat{\Sigma}_{\tilde{t}} + \lambda\mathbb{I}_d)^{-1}U(\hat{S}_{\tilde{t}} + \lambda\mathbb{I}_r) = \mathbb{I}_r \quad .$$

Note that

$$
\begin{aligned}
(\hat{S}_{\tilde{t}} + \lambda\mathbb{I}_r)U^{\top}(\hat{\Sigma}_{\tilde{t}} + \lambda\mathbb{I}_d)^{-1}U &= \left(U^{\top}\hat{\Sigma}_{\tilde{t}}U + \lambda\mathbb{I}_r\right)U^{\top}(\hat{\Sigma}_{\tilde{t}} + \lambda\mathbb{I}_d)^{-1}U \\
&= U^{\top}\left(\hat{\Sigma}_{\tilde{t}}UU^{\top} + \lambda\mathbb{I}_d\right)\left(\hat{\Sigma}_{\tilde{t}} + \lambda\mathbb{I}_d\right)^{-1}U \\
&= U^{\top}\left(\hat{\Sigma}_{\tilde{t}} + \lambda\mathbb{I}_d\right)\left(\hat{\Sigma}_{\tilde{t}} + \lambda\mathbb{I}_d\right)^{-1}U = \mathbb{I}_r \quad .
\end{aligned}
$$

With similar steps one can show that $U^{\top}(\hat{\Sigma}_{\tilde{t}} + \lambda\mathbb{I}_d)^{-1}U(\hat{S}_{\tilde{t}} + \lambda\mathbb{I}_r) = \mathbb{I}_r$, and therefore $U^{\top}(\hat{\Sigma}_{\tilde{t}} + \lambda\mathbb{I}_d)^{-1}U = (\hat{S}_{\tilde{t}} + \lambda\mathbb{I}_r)^{-1}$. By exploiting this fact, we can write

$$
\begin{aligned}
\|x\|_{V_{\tilde{t}}^{-1}}^2 &= \|x\|_2^2 \left(\frac{x}{\|x\|_2}^{\top}(\hat{\Sigma}_{\tilde{t}} + \lambda\mathbb{I}_d)^{-1}\frac{x}{\|x\|_2}\right) \\
&= \|x\|_2^2 \left(\frac{x}{\|x\|_2}^{\top}UU^{\top}(\hat{\Sigma}_{\tilde{t}} + \lambda\mathbb{I}_d)^{-1}UU^{\top}\frac{x}{\|x\|_2}\right) \\
&= \|x\|_2^2 \left(\frac{x}{\|x\|_2}^{\top}U(\hat{S}_{\tilde{t}} + \lambda\mathbb{I}_r)^{-1}U^{\top}\frac{x}{\|x\|_2}\right) \\
&= \|x\|_2^2 \left(\frac{x^{\top}U}{\|x^{\top}U\|_2}(\hat{S}_{\tilde{t}} + \lambda\mathbb{I}_r)^{-1}\frac{U^{\top}x}{\|U^{\top}x\|_2}\right) \quad ,
\end{aligned}
$$

where the second and last equations are results of Lemma A.3, and consequently

$$\|x\|_{V_{\tilde{t}}^{-1}}^2 \leq \frac{L^2}{\lambda_{\min}(\hat{S}_{\tilde{t}})} \quad . \tag{10}$$

On the other hand, from Lemma C.2, with probability at least $1 - \frac{\delta}{2}$, we have

$$\lambda_{\min}\left(\hat{S}_{t_0}\right) \geq \frac{(\tilde{t} - \tau_3)\lambda_{\min}^{+}(\Sigma)}{4} \quad . \tag{11}$$

Finally, by combining (10) and (11) with probability at least $1 - \frac{\delta}{2}$ we have

$$\|x\|_{V_{\tilde{t}}^{-1}}^2 \leq \frac{4L^2}{\lambda_{\min}^{+}(\Sigma)(\tilde{t} - \tau_3)} \quad .$$

$\square$

**Lemma C.4.** *With probability at least $1 - \frac{\delta}{4}$, for all $t \geq 3$ we have*

$$\|\mu^* - \mu_{\tilde{t}}\|_{V_{\tilde{t}}} \leq (\lambda^{\frac{1}{2}} + R + L)\sqrt{d\log((8 + 8\tilde{t}\max(L^2/\lambda, 1))/\delta)} + \lambda^{\frac{1}{2}}\|\mu^*\|_2 \quad .$$

*Proof.* Recall that by the definition of Alg. 1, we have $\mu_{\tilde{t}} = V_{\tilde{t}}^{-1} X_{1:\tilde{t}}^{\top} r_{1:\tilde{t}} + (1/d\sqrt{\tilde{t}}) \cdot \gamma_{\tilde{t}}$. Therefore, we can write

$$\|\mu^* - \mu_{\tilde{t}}\|_{V_{\tilde{t}}} \leq \underbrace{\|\mu^* - V_{\tilde{t}}^{-1} X_{1:\tilde{t}}^{\top} r_{1:\tilde{t}}\|_{V_{\tilde{t}}}}_{(I)} + \underbrace{\frac{\rho}{d\sqrt{\tilde{t}}}\|\gamma_{\tilde{t}}\|_{V_{\tilde{t}}}}_{(II)} .$$

We proceed the proof by providing upper bounds for (I) and (II). For (I), by invoking [1, Theroem 2], with probability at least $1 - \frac{\delta}{8}$, for all $t \geq 3$, which implies $\tilde{t} \geq 1$ we have

$$(I) \leq R\sqrt{d\log((8 + 8\tilde{t}L^2/\lambda)/\delta)} + \lambda^{\frac{1}{2}}\|\mu^*\|_2$$
$$\leq R\sqrt{d\log((8 + 8\tilde{t}\max(L^2/\lambda, 1))/\delta)} + \lambda^{\frac{1}{2}}\|\mu^*\|_2 .$$

On the other hand, since $\rho \leq 1$, for term (II) we have

$$(II) \leq \frac{1}{d\sqrt{\tilde{t}}}\|V_{\tilde{t}}\|_{\text{op}}^{\frac{1}{2}}\|\gamma_{\tilde{t}}\|_2 \leq \frac{L + \lambda^{\frac{1}{2}}}{d}\|\gamma_{\tilde{t}}\|_2 .$$

$$\mathbb{P}\big((II) \geq (L + \lambda^{\frac{1}{2}})\sqrt{\log(8d/\delta)}\big) \leq \mathbb{P}\big(\|\gamma_{\tilde{t}}\|_2 \geq d\sqrt{\log(8d/\delta)}\big)$$
$$\leq d\mathbb{P}\big(|\gamma_{1,\tilde{t}}| \geq \sqrt{\log(8d/\delta)}\big) \leq \frac{\delta}{8} .$$

Thus, by applying the union bound with probability at least $1 - \frac{\delta}{8}$, for all $t \geq 3$ we have

$$(II) \leq (L + \lambda^{\frac{1}{2}})\sqrt{\log(8\tilde{t}d/\delta)} \leq (L + \lambda^{\frac{1}{2}})\sqrt{d\log((8 + 8\tilde{t}\max(L^2/\lambda, 1))/\delta)} .$$

$\square$

*Proof of Lemma 5.3.* Recall that $\tau_3 = \max(\tau_1, \tau_2)$. From Lemma C.3, with probability at least $1 - \frac{\delta}{2}$ for all $t \geq 2\tau_3 + 3$ we have

$$\|x\|_{V_{\tilde{t}}^{-1}} \leq \frac{2L}{\sqrt{\lambda_{\min}^+(\Sigma)(\tilde{t} - \tau_3)}} .$$

From Lemma C.4, with probability at least $1 - \frac{\delta}{4}$ for all $t \geq 3$

$$\|\mu^* - \mu_{\tilde{t}}\|_{V_{\tilde{t}}} \leq (\lambda^{\frac{1}{2}} + R + L)\sqrt{d\log((8 + 8\tilde{t}\max(L^2/\lambda, 1))/\delta)} + \lambda^{\frac{1}{2}}\|\mu^*\|_2 .$$

Thus, combining Lemmas C.3 and C.4, with probability at least $1 - \frac{3\delta}{4}$ for all $t \geq 2\tau_3 + 3$ we have

$$\|\mu^* - \mu_{\tilde{t}}\|_{V_{\tilde{t}}}\|x\|_{V_t^{-1}} \leq$$
$$\frac{2L}{\sqrt{\lambda_{\min}^+(\Sigma)(\tilde{t} - \tau_3)}}\left((\lambda^{\frac{1}{2}} + R + L)\sqrt{d\log((8 + 8\tilde{t}\max(L^2/\lambda, 1))/\delta)} + \lambda^{\frac{1}{2}}\|\mu^*\|_2\right) .$$

By the fact that $t \geq 4\tau_3 + 3$, we have $\tilde{t} \geq 2\tau_3$, which implies $\frac{1}{\sqrt{\tilde{t} - \tau_3}} \leq \sqrt{\frac{2}{t}}$. We conclude the proof by using the inequality $\tilde{t} \geq \frac{t-3}{2} \geq \frac{t}{8}$, for all $t \geq 4$. $\square$

### C.4 Proof of Theorem 1

*Proof.* Combining Lemma 5.2 with Lemma 5.3 and using $1/(t-1) \leq 3/(4t)$ for all $t \geq 4$ we obtain, with probability at least $1 - \delta$ and for all $\tau \leq t \leq T$

$$\mathcal{F}_{a_t^*}(\langle\mu^*, X_{t,a_t^*}\rangle) - \mathcal{F}_{a_t}(\langle\mu^*, X_{t,a_t}\rangle) \leq 4\sqrt{\frac{\log(8KT/\delta)}{3t}}$$
$$+ \frac{48ML}{\sqrt{\lambda_{\min}^+(\Sigma)t}}\left((\lambda^{\frac{1}{2}} + R + L)\sqrt{d\log((8 + 4t\max(L^2/\lambda, 1))/\delta)} + \lambda^{\frac{1}{2}}\|\mu^*\|_2\right) .$$

By summing up the last inequality, with probability at least $1 - \delta$ we get

$$\sum_{t=\tau}^{T} \left[ \mathcal{F}_{a_t^*}(\langle \mu^*, X_{t,a_t^*} \rangle) - \mathcal{F}_{a_t}(\langle \mu^*, X_{t,a_t} \rangle) \right] \leq 8\sqrt{\frac{T \log(8KT/\delta)}{3}}$$
$$+ \frac{96ML}{\sqrt{\lambda_{\min}^+(\Sigma)}} \left( (\lambda^{\frac{1}{2}} + R + L)\sqrt{dT \log((8 + 4T \max(L^2/\lambda, 1))/\delta)} + \sqrt{\lambda T}\|\mu^*\|_2 \right).$$

(12)

where the last display is obtained by the inequality $\sum_{t=1}^{T} t^{-\frac{1}{2}} \leq 2T^{\frac{1}{2}}$. On the other hand, for $t \in [T]$, $\mathcal{F}_{a_t^*}(\langle \mu^*, X_{t,a_t^*} \rangle) - \mathcal{F}_{a_t}(\langle \mu^*, X_{t,a_t} \rangle) \leq 1$, and we can write

$$\sum_{t=1}^{\tau} \left[ \mathcal{F}_{a_t^*}(\langle \mu^*, X_{t,a_t^*} \rangle) - \mathcal{F}_{a_t}(\langle \mu^*, X_{t,a_t} \rangle) \right] \leq \tau .$$

(13)

By combining (12) and (13), we conclude the proof. □

# D   Experiments

In this section we include additional details on the simulation experiments in Sec. 6 and an experiment on the US census data.

## D.1   Additional Details on the Simulation

We use the following value for the underlying linear model used in Fig. 1.

$$\mu^* = (\underbrace{4, 3, 7, 0}_{\text{Group 1}}, \underbrace{8, 0, 0, 0}_{\text{Group 2}}, \underbrace{5, 5, 0, 0}_{\text{Group 3}}, \underbrace{2, 2, 2, 2}_{\text{Group 4}}, 1) .$$

Each slice of 4 coordinates of $\mu^*$ affects a different group. Furthermore, since each coordinate of $Y_a$ follows a standard uniform distribution, the resulting reward distributions for each group follow weighted variants of the Irwin-Hall distribution [18].

## D.2   Experiments on US Census data

In this section, we present an experiment performed using the US Census data and the FalkTables library[4] [13]. In particular we construct a dataset with features similar to the UCI Adult dataset but where the target is the person's income instead of the binary variable indicating if the income is more or less than $50K$ dollars. We use this target as a possibly inaccurate proxy for how well a candidate will perform on the job, hence it is used as the noisy reward for the bandit problem.

**Setup and Preprocessing.** To setup the bandit problem, we construct 2 datasets, namely $D1$ and $D2$, by selecting $500K$ males and $500K$ females random samples first from the 2017 US Census Survey, to assemble $D1$, and then from the 2018 survey to assemble $D2$. We use $D1$ to find mean and standard deviation for each feature and also for the target. After that we normalize features and target from $D2$ by subtracting the mean and dividing by the standard deviation previously computed on $D1$. We then construct $\mu^*$ as a ridge regression estimate on the samples from $D2$ with the regularization parameter equal to $10^{-8}$. The regression vector $\mu^*$ will be used to compute the (true) rewards for the samples. We construct the bandit problem with $K = 2$ arms/groups which correspond to the gender identities male and female. At each round, the context vectors of one male and one female candidate are sampled from $D2$ and after one of the two is selected by the policy, its corresponding noisy reward (i.e. its income) is received by the agent.

**Baselines.** We compare our method, namely *Fair-greedy* (Alg. 1), with the following baselines.

- *Uniform Random*, which selects an arm uniformly at random at each round.
- *OFUL* [1], with exploration parameter set to $0.1$.

---

[4]`https://github.com/zykls/folktables`

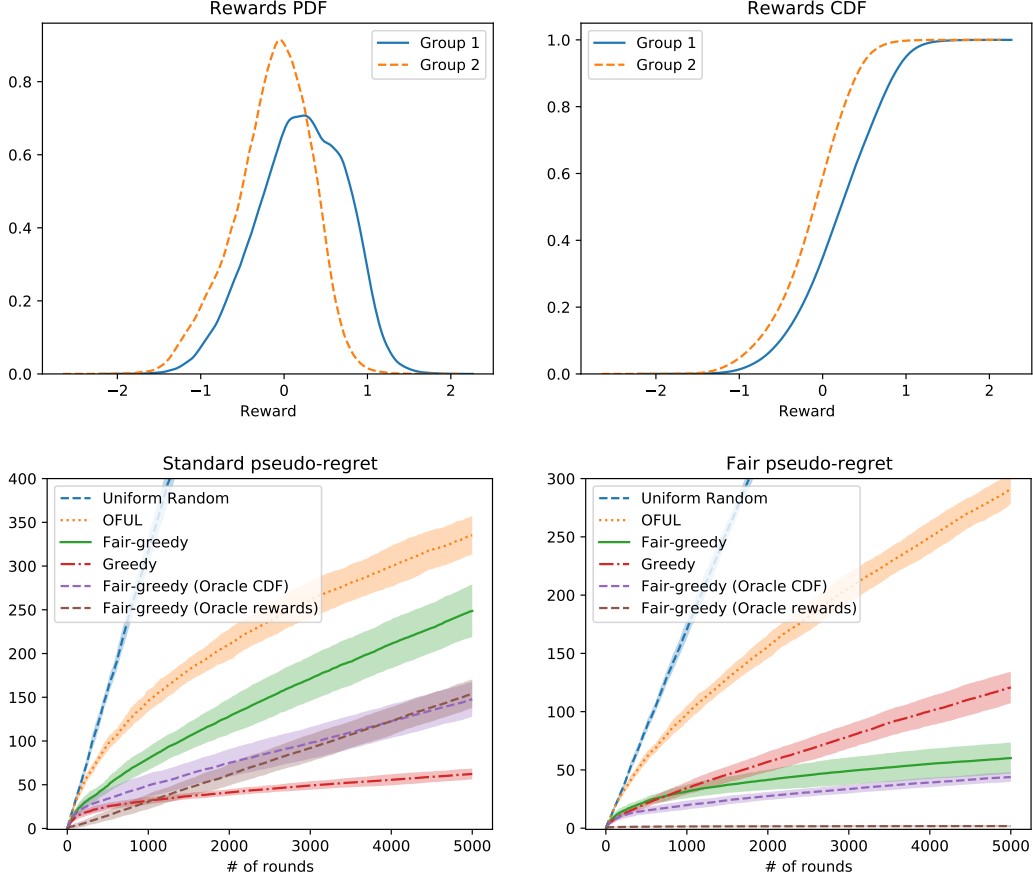

Figure 3: **US Census Results**. Top two images are density and CDF plots of the reward distributions while the bottom two plots are the standard and fair pseudo-regrets, with mean (solid lines) $\pm$ standard deviation (shaded region) over 10 runs. To compute the reward PDF and CDF for each group we use the empirical CDF on all $500K$ samples from $D2$.

- *Greedy*, which computes the ridge regression estimate for the reward vector using all the selected contexts and noisy rewards in the history and then selects the arm maximising the estimated reward.

- *Fair-greedy (Oracle CDF)*, which is a variant of *Fair-greedy* where all the selected contexts and noisy rewards in the history are used to compute the ridge regression estimate and the empirical CDF of each group is replaced by the true CDF.

- *Fair-greedy (Oracle rewards)*, which is another variant of *Fair-greedy* where the ridge regression estimate is replaced by the true reward model $\mu^*$ and all contexts in the history are used to compute the empirical CDF for each group.

Note that the last two methods are *oracle methods* because they rely either on the true CDF of the rewards for each group or on $\mu^*$, which are unknown to the agent. All methods using a ridge regression estimate have the regularization parameter set to $0.1$. We observed that varying this parameter did not affect much the relative performance of the methods.

**Results.** The results and the reward distributions are illustrated in Fig. 3. We note that in this case, *Greedy* performs much better than OFUL, which appears to be too conservative for this problem. In particular, the standard pseudo-regret of *Greedy* is unrivaled after 1000 rounds. Furthermore, since there is a large overlap in the distributions of rewards, our *Fair-greedy* policy performs much better than the *Uniform Random* policy even in terms of standard pseudo-regret, while it outperforms all non-oracle methods in terms of fair pseudo-regret. As expected, the oracle methods both achieve a

lower fair pseudo-regret than *Fair-greedy*, and we note that knowing only the underlying model $\mu^*$ is significantly more advantageous than knowing only the CDF for each group.

## E    Multiple Candidates for Each Group

This section contains a rigorous treatement of the content in Sec. 7. We consider the more realistic case where contexts from a given arm do not necessarily belong to the same group. In particular, we assume that at each round $t$, the agent receives tuples $\{(X_{t,a}, s_{t,a})\}_{a=1}^K$, where $s_{t,a} \in [G]$ is the sensitive group of the context $X_{t,a} \in \mathbb{R}^d$ and $G$ is the total number of groups. After that the agent selects action $a_t$ and subsequently receives the noisy reward $\langle \mu^*, X_{t,a} \rangle + \eta_t$

Note that we recover the original setting discussed in Sec. 3 when $G = K$ and $s_{t,a} = a$ for every $a \in [K]$, $t \in \mathbb{N}$. A more realistic scenario is when $\{(X_{t,a}, s_{t,a})\}_{a=1}^K$ are i.i.d., and the distribution represents e.g. the underlying population of candidates, where $\mathbb{P}(s_{t,a} = i)$ is the same for all $a \in [K]$ and can be small when the group $i$ is a minority. The following analysis applies to both cases.

We impose the following assumption, which is a natural extension of Assumption A.

**Assumption B.** *Let $\mu^* \in \mathbb{R}^d$ be the underlying reward model. We assume that:*

(i) *The noise random variable $\eta_t$ is zero mean $R$-subgaussian, conditioned on $\mathcal{H}_{t-1}$.*

(ii) *For any $a \in [K]$, let $(X_a, s_a)$ be a random variable with values in $\mathbb{R}^d \times [G]$ and $\|X_a\|_2 \leq L$ almost surely. $\{(X_{t,a}, s_{t,a})\}_{t=1}^T$ are i.i.d. copies of $(X_a, s_a)$. $X_a$ conditioned to $s_a = i$ is a copy of the random variable $\hat{X}_i$ which is independent on the arm, for every $a \in [K]$.*

(iii) *For every $a \in [K]$ $X_a$ conditioned to $s_a$ is independent from $(X_{a'}, s_{a'})$ for any $a' \neq a$.*

(iv) *For every $i \in [G]$, then there exist $d_i \geq 1$, an absolutely continuous random variable $Y_i$ with values in $\mathbb{R}^{d_i}$ admitting a density $f_i$, $B_i \in \mathbb{R}^{d \times d_i}$ and $c_i \in \mathbb{R}^d$ such that $B_i^\top B_i = \mathbb{I}_{d_i}$,*

$$\hat{X}_i = B_i Y_i + c_i \quad and \quad \mu^{*\top} B_i \neq 0 \ .$$

We define $\mathcal{F}(r, i) = \mathbb{P}(\langle \mu^*, \hat{X}_i \rangle \leq r) = \mathbb{P}(\langle \mu^*, X_a \rangle \leq r \,|\, s_a = i)$ for any $r \in \mathbb{R}$, $i \in G$. Hence we can extend the definition of group meritocratic fairness as follows.

**Definition E.1** (GMF policy). *a policy $\{a_t^*\}_{t=1}^\infty$ is group meritocratic fair (GMF) if for all $t \in \mathbb{N}, a \in [K]$ it satisfies*

$$\mathcal{F}(\langle \mu^*, X_{t,a_t^*} \rangle, s_{a_t^*}) \geq \mathcal{F}(\langle \mu^*, X_{t,a} \rangle, s_{a_t}) \ .$$

The fair pseudo-regret is now defined as

$$R_F(T) = \sum_{t=1}^T \mathcal{F}(\langle \mu^*, X_{t,a_t^*} \rangle, s_{a_t^*}) - \mathcal{F}(\langle \mu^*, X_{t,a_t} \rangle, s_{a_t}) \tag{14}$$

We can adapt Proposition 3.1 to this setting as follows.

**Proposition E.1** (GMF policy satisfies *history-agnostic demographic parity*). *Let $\{\langle \mu^*, X_a \rangle\}_{a=1}^K$ conditioned to $\{s_a\}_{a=1}^K$ be independent and absolutely continuous and for every $a \in [K], t \in \mathbb{N}$, let $(X_{t,a}, s_{t,a})$ be an i.i.d. copy of $(X_a, s_a)$. Then for every $t \in \mathbb{N}$, $\{\mathcal{F}(\langle \mu^*, X_{t,a} \rangle, s_{t,a})\}_{a=1}^K$ conditioned to $\{s_{t,a}\}_{a=1}^K$ are i.i.d. uniform on $[0,1]$ and*

$$\mathbb{P}(a_t^* = a \,|\, s_{t,a}, \mathcal{H}_{t-1}^-) = 1/K \qquad \forall a \in [K], \tag{15}$$

*for any GMF policy $\{a_t^*\}_{t=1}^\infty$.*

*Proof.* Let $\psi_a := \mathcal{F}(\langle \mu^*, X_{t,a} \rangle, s_{t,a})$. From the assumptions $\{\psi_a\}_{a=1}^K$ conditioned to $\{s_{t,a}\}_{a=1}^K$ are i.i.d random variables, independent from $\mathcal{H}_{t-1}^-$, with uniform distribution on $[0,1]$ (see [9, Theorem 2.1.10]). Let $\tilde{\mathbb{P}} = \mathbb{P}(\cdot \,|\, \{s_{t,a}\}_{a=1}^K, \mathcal{H}_{t-1}^-)$, we have that $\forall a_1, a_2 \in [K]$: $\tilde{\mathbb{P}}(\psi_{a_1} = \psi_{a_2}) = 0$, $\tilde{\mathbb{P}}(a_t^* = a \,|\, \mathcal{H}_{t-1}^-) = \tilde{\mathbb{P}}(a_t^* = a)$ and

$$\tilde{\mathbb{P}}(a_t^* = a_1) = \tilde{\mathbb{P}}(\psi_{a_1} > \psi_{a'}, \forall a' \neq a_1) = \tilde{\mathbb{P}}(\psi_{a_2} > \psi_{a'}, \forall a' \neq a_2) = \tilde{\mathbb{P}}(a_t^* = a_2) = 1/K \ .$$

Let $S_{t,\not a} = \{s_{t,a} : a \in [K]/a\}$, then the statement follows from

$$\mathbb{P}(a_t^* = a \mid s_{t,a}, \mathcal{H}_{t-1}^-) = \mathbb{E}_{S_{t,\not a}}[\tilde{\mathbb{P}}(a_t^* = a)] \ .$$

$\square$

Proposition E.1 states that the probability of selecting an arm does not change based on group membership. Fair-Greedy V2 in Alg. 2 is the extension of the fair-greedy policy to this new setting.

---

**Algorithm 2** Fair-Greedy V2

---

1: **Requires** regularization parameter $\lambda > 0$, and noise magnitude $\rho \in (0, 1]$
2: **for** $t = 1 \dots T$ **do**
3:     Receive $\{(X_{t,a}, s_{t,a})\}_{a=1}^K$
4:     Set $\tilde{t} = \lfloor (t-1)/2 \rfloor$, $X_{1:\tilde{t}} = (X_{1,a_1}, \dots, X_{\tilde{t},a_{\tilde{t}}})^\top$, $r_{1:\tilde{t}} = (r_{1,a_1}, \dots, r_{\tilde{t},a_{\tilde{t}}})$.
5:     **If** $\tilde{t} = 0$ set $\mu_{\tilde{t}} = 0$, **else** let $V_{\tilde{t}} := X_{1:\tilde{t}}^\top X_{1:\tilde{t}} + \lambda \mathbb{I}_d$, generate $\gamma_{\tilde{t}} \sim \mathcal{N}(0, \mathbb{I}_d)$ and compute

$$\mu_{\tilde{t}} := V_{\tilde{t}}^{-1} X_{1:\tilde{t}}^\top r_{1:\tilde{t}} + \frac{\rho}{d\sqrt{\tilde{t}}} \cdot \gamma_{\tilde{t}} \ .$$

6:     For each $a \in [K]$, let $i := s_{t,a}$ and $N_{t,i} = \sum_{j=\tilde{t}+1}^{t-1} \sum_{a'=1}^K \mathbb{1}\{s_{j,a'} = i\}$, compute

$$\hat{\mathcal{F}}_t(\langle \mu_{\tilde{t}}, X_{t,a} \rangle, i) := N_{t,i}^{-1} \sum_{j=\tilde{t}+1}^{t-1} \sum_{a'=1}^K \mathbb{1}\{\langle \mu_{\tilde{t}}, X_{j,a'} \rangle \leq \langle \mu_{\tilde{t}}, X_{t,a} \rangle\} \mathbb{1}\{s_{j,a'} = i\} \ .$$

7:     Sample action

$$a_t \sim \mathcal{U}\Big[\underset{a \in [K]}{\arg\max} \, \hat{\mathcal{F}}_t(\langle \mu_{\tilde{t}}, X_{t,a} \rangle, s_{t,a})\Big] \ .$$

8:     Observe noisy reward $r_{t,a_t} = \langle \mu, X_{t,a_t} \rangle + \eta_t$.
9: **end for**

---

Notice that the number of contexts used for the CDF approximation for group $i \in [G]$ is now the random variable $N_{t,i}$. Furthermore, we are now using contexts from all the arms to estimate the CDFs, which as we will see it can improve the dependency on $K$ in the fair pseudo-regret bound. We observe that the information averaged demographic parity property of Lemma 4.1 does not transfer directly to Fair-Greedy V2, because at each round, there can be a different number of candidates for each group. However, as we will see, the regret is still similar to the original case.

The following Lemma establishes an high probability lower bound on $N_{t,i}$.

**Lemma E.1.** *Let $q_K := \min_{i \in [G]} \sum_{a=1}^K \mathbb{P}(s_a = i)$ and let*

$$\mathcal{R} = \mathbb{1}\{\exists a \in [K] \text{ such that } \forall i \in [G] \, \mathbb{P}(s_a = i) < 1\} \ .$$

*$\mathcal{R} = 1$ means that the sensitive attribute is random for at least one arm, while is deterministic if $\mathcal{R} = 0$. Then, let $\alpha = \mathcal{R}b + (1 - \mathcal{R})$ with $b \in (0, 1)$ and $t_N := 3 + \mathcal{R}\lceil \frac{2}{(1-\alpha)^2 q_K} \log(GT/\delta) \rceil$, with $N_{t,i}$ defined at Line 6 of Alg. 2 and, without loss of generality, $q_K > 0$. For simiplicity we let $\mathcal{R}x = 0$ when $\mathcal{R} = 0, x = \infty$. We have that with probablity at least $1 - \mathcal{R}\delta$, for every $t \in \{t_N, \dots, T\}$*

$$\min_{i \in [G]} N_{t,i} \geq (t - 1 - \tilde{t})\alpha q_K$$

*Proof.* If $\mathcal{R} = 0$, then $\mathbb{P}(s_a = i) = \mathbb{1}\{s_a = i\}$ and the result follows.

If $\mathcal{R} = 1$ instead, note that for every $i \in [G]$ we have that

$$\mathbb{E}[N_{t,i}] = \sum_{j=\tilde{t}-1}^{t-1} \sum_{a=1}^K \mathbb{P}(s_a = i) \geq (t - 1 - \tilde{t})q_K \ .$$

Applying the Chernoff bound we have that with probability at least $1 - \delta$, for all $t > t_N$

$$N_{t,i} \geq \alpha \mathbb{E}[N_{t,i}] \geq (t - 1 - \tilde{t})\alpha q_K \ ,$$

and the statement follows $\square$

Let $S_T(t_N, \alpha) := \left\{ \{\{s_{t,a}\}_{a=1}^K\}_{t=1}^T : \min_i N_{t,i} \geq (t - \tilde{t} - 1)\alpha q_K \text{ for all } t_N \leq t \leq T \right\}$ be the event when Lemma E.1 is satisfied. We can then proceed the analysis assuming that $S_T(t_N, \alpha)$ holds. Noticing that the maximum number of approximate CDFs to be computed at each round is $G$ we can adapt Lemma C.1 as follows.

**Lemma E.2.** *Let Assumption B*(ii) *hold and $\hat{\mathcal{F}}_t(r, i)$, and $\mu_{\tilde{t}}$ to be generated by Alg. 2. Let $Z_i := \langle \mu_{\tilde{t}}, \hat{X}_i \rangle$ and denote with $\mathcal{F}_{Z_i}(\cdot)$ its CDF, conditioned on $\mu_{\tilde{t}}$. Then, if the event $S_T(t_N, \alpha)$ is satisfied, with probability at least $1 - \delta$ we have that for all $t_N \leq t \leq T$*

$$\sup_{i \in [G], r \in \mathbb{R}} |\hat{\mathcal{F}}_t(r, i) - \mathcal{F}_{Z_i}(r)| \leq \sqrt{\frac{\log(2GT/\delta)}{(t-1)\alpha q_K}} \ .$$

Then, following the steps in Lemma 5.2, we obtain the following bound on the instantaneous regret.

**Lemma E.3** (Instant regret bound). *Let Assumption B*(ii)(iv) *hold and $a_t$ to be generated by Alg. 2. Then, if the event $S_T(t_N, \alpha)$ is satisfied, with probability at least $1 - \delta/4$, for all $t$ such that $t_N \leq t \leq T$ we have*

$$\mathcal{F}(\langle \mu^*, X_{t,a_t^*} \rangle, s_{a_t^*}) - \mathcal{F}(\langle \mu^*, X_{t,a_t} \rangle, s_{a_t}) \leq 6M\|\mu^* - \mu_{\tilde{t}}\|_{V_{\tilde{t}}} \|x_{\max}\|_{V_{\tilde{t}}^{-1}} + 2\sqrt{\frac{\log(16GT/\delta)}{(t-1)\alpha q_K}} \ ,$$

*where $\|x_{\max}\|_{V_{\tilde{t}}^{-1}} := \sup_{x \in \cup_{i=1}^G \text{Supp}(\hat{X}_i)} \|x\|_{V_{\tilde{t}}^{-1}}$.*

*Proof sketch.* Uses the decomposition in the proof of Lemma 5.2, then Lemma E.2 and a version of Lemma 5.1 adapted to this more general setting. $\square$

We can bound $\|\mu^* - \mu_{\tilde{t}}\|_{V_{\tilde{t}}}$ using the confidence bounds in OFUL [1]. To bound $\|x_{\max}\|_{V_{\tilde{t}}^{-1}}$ instead, we first provide an adaptation of Proposition 5.1, which guarantees sufficient exploration of all arms. The proof is very similar to that of Proposition 5.1 and we report it here for completeness.

**Proposition E.2.** *Let Assumption A hold, $a_t$ be generated by Alg. 1 and $c \in [0, 1)$. Then, if $S_T(t_N, \alpha)$ is satisfied, with probability at least $1 - \delta/4$, for all $a \in [K]$ and all $t \geq \max\left(t_N, 3 + 8\log^{3/2}\left(5G\,\mathrm{e}/\delta\right)\left(1 - \sqrt[K]{c}\right)^{-3}(q_K\alpha)^{-3/2}\right)$ we have*

$$\mathbb{P}(a_t = a \mid s_{t,a}, \mathcal{H}_{t-1}^-) \geq \frac{c}{K} \ ,$$

*where we recall that $\mathcal{H}_t^- = \cup_{i=1}^t \left\{ \{(X_{i,a}, s_{i,a})\}_{a=1}^K, r_{i,a_i}, a_i \right\}$.*

*Proof.* Recall the definition of Alg. 2. For any $a \in [K]$ let $\hat{r}_{t,a} = \mu_{\tilde{t}}^\top X_{t,a}$, which is the estimated reward for arm $a$, at round $t$. Note that $\mu_{\tilde{t}}$ and $X_{t,a}$ are independent random variables. Furthermore, denote with $\mathcal{F}_{\hat{r}_{t,a}}(\cdot, s_{t,a})$ the CDF of $\hat{r}_{t,a}$ conditioned on $\mu_{\tilde{t}}$ and $s_{t,a}$, and let

$$\phi_{t,a} := \mathcal{F}_{\hat{r}_{t,a}}(\hat{r}_{t,a}, s_{t,a}) \ , \quad \text{and} \quad \hat{\phi}_{t,a} := \hat{\mathcal{F}}_t(\hat{r}_{t,a}, s_{t,a}) \ .$$

Let $C_t := \operatorname{argmax}_{a \in [K]} \hat{\phi}_{t,a}$. Now, by the definition of the algorithm, we have

$$\mathbb{P}(a_t = a \mid \{s_{t,a}\}_{a=1}^K, \mathcal{H}_{t-1}^-) = \sum_{m=1}^K \frac{1}{m} \mathbb{P}(a \in C_t, |C_t| = m \mid \mathcal{H}_{t-1}^-) \ ,$$

Let $\epsilon_t > 0$ and $\tilde{\mathbb{P}}(\cdot) = \mathbb{P}(\cdot \mid \{s_{t,a}\}_{a=1}^K, \mathcal{H}_{t-1}^-, \sup_{a \in [K]} |\phi_{t,a} - \hat{\phi}_{t,a}| \leq \epsilon_t)$. Then, we can write

$$\tilde{\mathbb{P}}(a_t = a) \geq \tilde{\mathbb{P}}(\hat{\phi}_{t,a} > \hat{\phi}_{t,a'}, \forall a' \neq a) \geq \tilde{\mathbb{P}}(\phi_{t,a} > \phi_{t,a'} + 2\epsilon_t, \forall a' \neq a) \ ,$$

where in the first inequality we considered the case when $a \in C_t$ and $|C_t| = 1$. In the second inequality we considered the worst case scenario where $\hat{\phi}_{t,a} = \phi_{t,a} - \epsilon_t$ and $\hat{\phi}_{t,a'} = \phi_{t,a'} + \epsilon_t$. Recall that by the construction of the algorithm $\mu_{\tilde{t}} = V_{\tilde{t}}^{-1} X_{1:\tilde{t}}^\top r_{1:\tilde{t}} + (1/\sqrt{d\tilde{t}}) \cdot \gamma_{\tilde{t}}$. for all $i \in [G]$,

the additive noise $(1/\sqrt{d\tilde{t}})\gamma_{\tilde{t}}$ assures that $\mu_{\tilde{t}}^{\top} B_i \neq 0$, almost surely. Therefore, by Lemma A.1 $\hat{r}_{t,a} = \langle \mu_{\tilde{t}}, X_{t,a} \rangle$ conditioned on $\mu_{\tilde{t}}$ is absolutely continuous.

Assumption A(iii) and [9, Theorem 2.1.10] yield that $\{\phi_{t,a}\}_{a \in [K]}$ conditioned to $\{s_{t,a}\}_{a=1}^{K}$ are independent and uniformly distributed on $[0,1]$ and in turn that

$$\tilde{\mathbb{P}}(a_t = a) \geq \int_0^1 \left( \mathbb{P}(\phi_{t,a'} < \mu - 2\epsilon_t) \right)^{K-1} \mathrm{d}\mu = \int_{2\epsilon_t}^1 (\mu - 2\epsilon_t)^{K-1} \mathrm{d}\mu = \frac{(1-2\epsilon_t)^K}{K}. \qquad (16)$$

We continue by computing an $\epsilon_t$ for which $\sup_{a \in [K]} |\phi_{t,a} - \hat{\phi}_{t,a}| \leq \epsilon_t$ holds with high probability. Observing that, conditioned on $\mu_{\tilde{t}}$ and $\{s_{t,a}\}_{a=1}^{K}$, $\hat{\mathcal{F}}_{t,a}(\cdot, s_{t,a})$ is the empirical CDF of $\mathcal{F}_{\hat{r}_{t,a}}(\cdot, s_{t,a})$, we can use Lemma E.1 and the Dvoretzky–Kiefer–Wolfowitz-Massart inequality to obtain, for any $a \in [K]$, $t \geq t_N$, and $s \geq 0$

$$\mathbb{P}\left( |\phi_{t,a} - \hat{\phi}_{t,a}| \geq s \right) \leq 2\exp\left( -2s^2(t - \tilde{t} - 1)(\alpha q_K) \right) .$$

Now, let $\tau_0 := \max\left( t_N, 3 + 8\log^{3/2}\left(5G\,\mathrm{e}/\delta\right)\left(1 - \sqrt[K]{c}\right)^{-3}(\alpha q_K)^{-3/2} \right)$. By applying the union bound and noticing that we have max of $G$ CDFs and approximate CDFs, we can write

$$\mathbb{P}\left( \sup_{t \geq \tau_0, a \in [K]} |\phi_{t,a} - \hat{\phi}_{t,a}| \geq s \right) \leq G \sum_{t=\tau_0}^{\infty} \mathbb{P}\left( |\phi_{t,a} - \hat{\phi}_{t,a}| \geq s \right)$$

$$\leq 2G \sum_{t=\tau_0}^{\infty} \exp\left( -2s^2(t - \tilde{t} - 1)(\alpha q_K) \right).$$

Since $\tilde{t} = \lfloor \frac{t-1}{2} \rfloor$, it is straightforward to check that

$$\mathbb{P}\left( \sup_{t \geq \tau_0, a \in [K]} |\phi_{t,a} - \hat{\phi}_{t,a}| \geq s \right) \leq 2G \int_{t=\tau_0-1}^{\infty} \exp\left( -s^2 \alpha q_K t \right) \mathrm{d}t$$

$$\leq \frac{2G}{\alpha q_K s^2} \exp\left( -s^2 \alpha q_K (\tau_0 - 1) \right) .$$

Now, for any $\delta \in (0,1)$, by assigning $s = \sqrt{\frac{\log(4G(\tau_0-1)/\delta)}{(\tau_0-1)\alpha q_K}}$, we get

$$\mathbb{P}\left( \sup_{t \geq \tau_0, a \in [K]} |\phi_{t,a} - \hat{\phi}_{t,a}| \geq \sqrt{\frac{\log(4G(\tau_0-1)/\delta)}{(\tau_0-1)\alpha q_K}} \right) \leq \frac{\delta}{2\log\left(4G(\tau_0-1)/\delta\right)} \leq \frac{\delta}{4} , \qquad (17)$$

where from $\tau_0 \geq 3, \delta < 1 \implies 4G(\tau_0 - 1)/\delta \geq 8 \geq \mathrm{e}^2 \implies \log\left(4G(\tau_0 - 1)/\delta\right) \geq 2$ we obtain the last inequality. From (16), it follows that

$$\inf_{t \geq \tau_0, a \in [K]} \mathbb{P}\left( a_t = a | \{s_{t,a}\}_{a=1}^K, \mathcal{H}_{t-1}^- \right) \geq \frac{(1 - 2\sup_{t \geq \tau} \epsilon_t)^K}{K} .$$

Moreover, form (17), by letting $\epsilon_t = \sqrt{\frac{\log(4G(\tau_0-1)/\delta)}{(\tau_0-1)\alpha q_K}}$, with probability at least $1 - \frac{\delta}{4}$, we have

$$\inf_{t \geq \tau, a \in [K]} \mathbb{P}\left( a_t = a | \{s_{t,a}\}_{a=1}^K, \mathcal{H}_{t-1}^- \right) \geq \frac{1}{K}\left( 1 - 2\underbrace{\sqrt{\frac{\log(4G(\tau_0-1)/\delta)}{(\tau_0-1)\alpha q_K}}}_{(I)} \right)^K . \qquad (18)$$

For the term (I) in the above, using $\log(x) \leq \log(5\,\mathrm{e}/4)x^{1/3}$ and $x \geq x^{2/3}$ for any $x \geq 1$ we deduce that

$$(I) = \frac{\log(4G/\delta) + \log(\tau_0 - 1)}{(\tau_0 - 1)\alpha q_K} \leq \frac{\log(4G/\delta) + \log(5\,\mathrm{e}/4)}{(\tau_0 - 1)^{2/3}\alpha q_K} = \frac{\log(5G\,\mathrm{e}/\delta)}{(\tau_0 - 1)^{2/3}\alpha q_K} .$$

Now, since $\tau_0 \geq 3 + 8\log^{3/2}\left(5G\,\mathrm{e}/\delta\right)\left(1 - \sqrt[K]{c}\right)^{-3}(\alpha q_K)^{-3/2}$, we get that $(I) \leq \frac{1}{4}\left(1 - \sqrt[K]{c}\right)^2$ and conclude the proof by plugging this inequality in (18). $\qquad \square$

Furthermore, for fixed $t$, let $\tilde{\mathbb{E}} = \mathbb{E}[\cdot \mid \mathcal{H}_{t-1}^-]$ and $\tilde{\mathbb{P}} = \mathbb{P}[\cdot \mid \mathcal{H}_{t-1}^-]$. Note that if the assumptions of Proposition E.2 are satisfied, then

$$
\begin{aligned}
\tilde{\mathbb{E}}[X_{t,a_t} X_{t,a_t}^\top] &= \sum_{i=1}^{G} \mathbb{E}[\hat{X}_i \hat{X}_i^\top] \tilde{\mathbb{P}}(s_{t,a_t} = i) \\
&= \sum_{i=1}^{G} \mathbb{E}[\hat{X}_i \hat{X}_i^\top] \sum_{a=1}^{K} \tilde{\mathbb{P}}(a_t = a \mid s_{t,a} = i) \tilde{\mathbb{P}}(s_{t,a} = i) \\
&\geq cK^{-1} \sum_{i=1}^{G} \mathbb{E}[\hat{X}_i \hat{X}_i^\top] q_K = c \frac{q_K G}{K} \frac{1}{G} \sum_{i=1}^{G} \mathbb{E}[\hat{X}_i \hat{X}_i^\top] \qquad (19)
\end{aligned}
$$

where we applied Proposition E.2 in the last line. We can bound $\|x_{\max}\|_{V_t^{-1}}$ in Lemma E.3 in the same way as in Lemma C.3 using (19) with $c = 1/2$ when needed in the proof of Lemma C.2. Combining the previous results we obtain the following regret bound.

**Theorem 2.** *Let Assumption B hold, $a_t$ be generated by Alg. 2 and $\Sigma := G^{-1} \sum_{i=1}^{G} \mathbb{E}[\hat{X}_i \hat{X}_i^\top]$ Then, with probability at least $1 - \delta$, for any $T \geq 1$ we have*

$$
\begin{aligned}
R_F(T) \leq & \frac{96ML\sqrt{K}}{\sqrt{\lambda_{\min}^+(\Sigma) q_K G}} \left[ (\lambda^{\frac{1}{2}} + R + L)\sqrt{dT \log((8 + 4T \max(L^2/\lambda, 1))/\delta_1)} + \sqrt{\lambda T}\|\mu^*\|_2 \right] \\
& + 8\sqrt{\frac{T \log(8GT/\delta_1)}{3\alpha q_K}} + \tau ,
\end{aligned}
$$

*where $\delta = \delta_1 + \mathcal{R}\delta_2$, $\tau = 4\max(\tau_1, \tau_2, \mathcal{R}\tau_3) + 3$ and*

$$
\tau_1 = \frac{32K^3}{(\alpha q_K)^{3/2}} \log^{3/2}(5G\,\mathrm{e}/\delta_1), \quad \tau_2 = \frac{54L^2}{\lambda_{\min}^+(\Sigma)} \log(4d/\delta_1), \quad \tau_3 = \frac{2}{(1-\alpha)^2 q_K} \log(GT/\delta_2),
$$

*where $q_K$, $\mathcal{R}$ and $\alpha$ are defined in Lemma E.1 and we use the convention $\mathcal{R}\tau_3 = 0$ if $\mathcal{R} = 0$, $\tau_3 = \infty$. Hence*

$$
R_F(T) = O\left( \mathcal{R}\frac{\log(GT/\delta_2)}{q_K} + \frac{K^3 \log^{3/2}(G/\delta_1)}{q_K^{3/2}} + \sqrt{\frac{dT \log(GT/\delta_1)}{(1 + G/K)q_K}} \right) .
$$

*Proof Sketch.* First, assume $S_T(t_N, \alpha)$ holds and use a similar strategy of Thm. 1 to get a bound w.p. at least $1 - \delta_1$. Then combine this result with Lemma E.1. $\qquad\square$

Notice that in the case where each arm corresponds to a different sensitive group, i.e. when $G = K$, $s_a = a$ and therefore $q_K = 1$, $\mathcal{R} = 0$ and $\alpha = 1$, we recover Thm. 1. Moreover, we have the following corollary which shows an advantage for higher number of arms compared to the bound in Thm. 1 when $\{(X_a, s_a)\}_{a=1}^{K}$ are i.i.d..

**Corollary E.1.** *Let $\{(X_a, s_a)\}_{a=1}^{K}$ be i.i.d. and $q_{\min} := \min_{i \in [G]} \mathbb{P}(s_a = i)G$. If Assumption B holds and $a_t$ is generated via Alg. 2 we have that with probability at least $1 - \delta$, for any $T \geq 1$ and $\alpha \in (0, 1)$ we have that*

$$
\begin{aligned}
R_F(T) \leq & \frac{96ML}{\sqrt{\lambda_{\min}^+(\Sigma) q_{\min}}} \left[ (\lambda^{\frac{1}{2}} + R + L)\sqrt{dT \log((8 + 4T \max(L^2/\lambda, 1))/\delta_1)} + \sqrt{\lambda T}\|\mu^*\|_2 \right] \\
& + 8\sqrt{\frac{TG \log(8GT/\delta_1)}{3K\alpha q_{\min}}} + \tau ,
\end{aligned}
$$

*where $\delta = \delta_1 + \delta_2$, $\tau = 4\max(\tau_1, \tau_2, \tau_3) + 3$ and*

$$
\tau_1 = \frac{32(KG)^{3/2}}{(\alpha q_{\min})^{3/2}} \log^{3/2}\left(\frac{5G\,\mathrm{e}}{\delta_1}\right), \quad \tau_2 = \frac{54L^2}{\lambda_{\min}^+(\Sigma)} \log\left(\frac{4d}{\delta_1}\right), \quad \tau_3 = \frac{2G}{(1-\alpha)^2 K q_{\min}} \log\left(\frac{GT}{\delta_2}\right).
$$

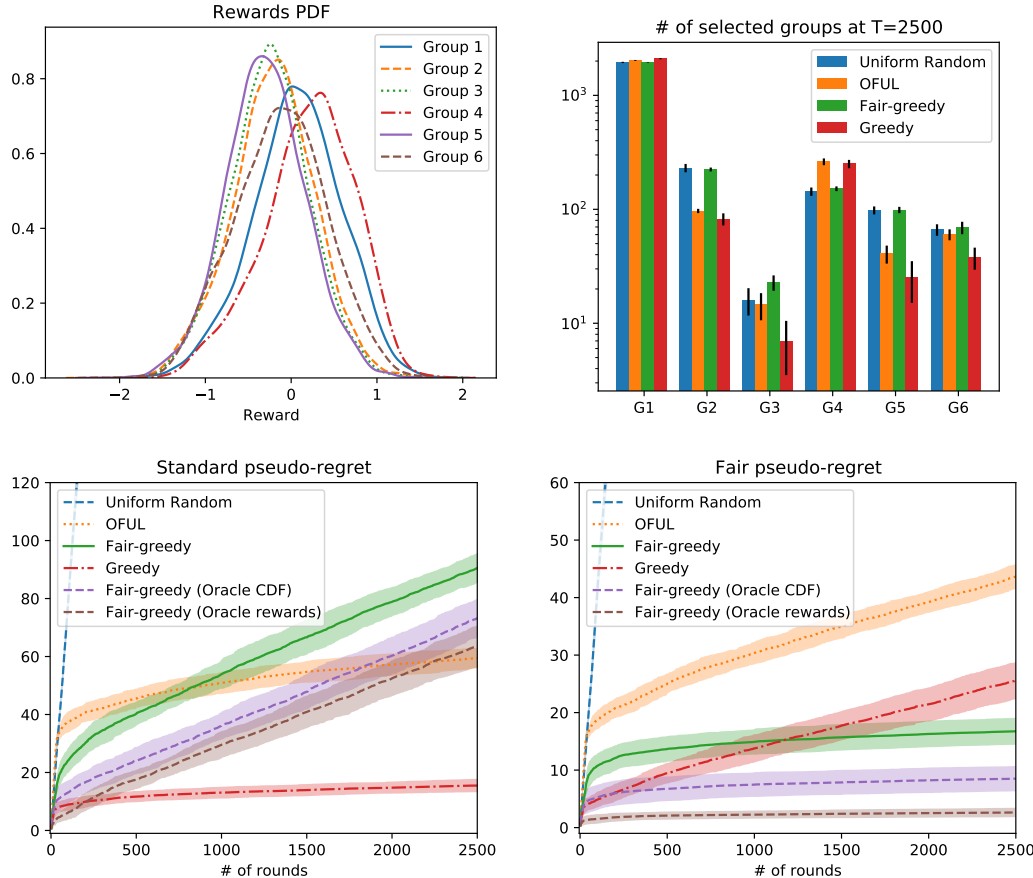

Figure 4: **US Census Results. Group = Ethnicity**. First image is the density plots of the reward distributions, the second image is the number candidates (in log scale) from each group which are selected by each policy (mean and std over 10 runs), while the bottom two plots are the standard and fair pseudo-regrets, with mean (solid lines) ± standard deviation (shaded region) over 10 runs. To compute the reward CDF for each group we use the empirical CDF on $5K$ samples from $D2$.

*Hence*

$$R_F(T) = O\left(\frac{G\log(GT/\delta)}{Kq_{\min}} + \frac{(KG)^{3/2}\log^{3/2}(G/\delta)}{q_{\min}^{3/2}} + \sqrt{\frac{dT\log(GT/\delta)}{(1+K/G)q_{\min}}}\right)$$

Note that in Cor. E.1, $q_{\min} > 0$ without loss of generality and $q_{\min} = 1$ if and only if each group has the same probability of being sampled. Furthermore $q_{\min}/G$ is the probability that a context belongs to the less common group, which depends on the problem at hand. Note that there is an advantage compared to Thm. 1 in terms of number of arms when $K > G$. This is because context coming from all arms can be use to estimate the CDF of a given group.

### E.1  Additional Details on the US Census Experiments

This experiment is introduced in Sec. 7 and similarly to that of Appendix D.2, is performed using the US Census data. However, candidates are sampled from the original dataset at random together with their sensitive group (their ethnicity). Hence, we use Fair-greedy V2 (Alg. 2). Differently from Appendix D.2 where we use the target income as noisy reward, here we add artificial noise with standard deviation 0.2 directly to the true reward.

**Setup and Preprocessing.** To setup the bandit problem, we construct two datasets: $D1$ and $D2$. We first load all the data from the 2017 US Census Survey to assemble $D1$, and then from the 2018

survey to assemble $D2$. Then we retain only candidates from 6 ethnic groups containing at least $5 \times 10^3$ candidates, in order to accurately compute the true CDF for each group. We use $D1$ to find mean and standard deviation for each feature and also for the target. After that we normalize features and target of $D2$ by subtracting the mean and dividing by the standard deviation previously computed on $D1$. We then construct $\mu^*$ as a ridge regression estimate on the samples from $D2$ with the regularization parameter equal to $10^{-8}$. The regression vector $\mu^*$ will be used to compute the (true) rewards for the samples. We construct the bandit problem as follows. At each round, the context vectors of $K = 10$ individual are sampled from $D2$ and after one is selected by the policy, its corresponding noisy reward, obtained by adding gaussian noise with standard deviation $0.2$ to the true reward, is received by the agent.

**Baselines.** We compare our method with the same baselines as in Appendix D.2, where the two oracle policies are now variants of Fair-Greedy V2. Moreover, we set the regularization parameter for all policies using a ridge estimate to $0.1$ and the exploration parameter of OFUL to $0.01$.

**Results (Fig. 4).** We draw similar conclusions as in Appendix D.2. In particular, Greedy performing better than OFUL and the Fair-Greedy policy achieving sublinear fair pseudo-regret, but worse than Oracle methods. Additionaly we can see that knowing $\mu^*$ plays a more important role than knowing the true reward CDFs. In this case, the gap between the Uniform random policy and the others is even larger since $K = 10$. Moreover, as expected, Fair-greedy selects more candidates from underperforming (in terms of reward) minority groups, when compared with OFUL and Greedy.

# F    Trade off between Fairness and Reward Maximization

In this section, we show for which problems the GMF policy and the optimal policy have competing goals. in particular, for the case of $K = 2$, when the rewards are absolutely continuous and independent across arms, whenever they are not identically distributed, the GMF policy achieves linear standard pseudo-regret with nonzero probability. The following theorem proves this result.

**Theorem 3.** *Let Assumption A hold with $K = 2$, and assume that $\mathcal{F}_1 \neq \mathcal{F}_2$. Let $\bar{r}_{t,a} := \langle \mu^*, X_{t,a} \rangle$, $\{a_t^*\}_{t=1}^T$ be the GMF policy (see Definition 3.1) and $\{a_t^{opt}\}_{t=1}^T$ be the optimal policy (see Remark 3.1). Then, there exists $\epsilon > 0$, such that*

$$p = \int_0^1 \left[ \max(\mathcal{F}_1(\mathcal{F}_2^{-1}(y) - \epsilon) - y, 0) + \max(y - \mathcal{F}_1(\mathcal{F}_2^{-1}(y) + \epsilon), 0) \right] \, \mathrm{d}y > 0 \ .$$

*Furthermore with probability at least $\frac{\epsilon p}{4L\|\mu^*\|}$, for any $T > 0$, we have*

$$T \cdot \frac{\epsilon p}{2} \leq \sum_{t=1}^T \left[ \bar{r}_{t,a_t^{opt}} - \bar{r}_{t,a_t^*} \right] \ . \tag{20}$$

*Proof.* Let $\bar{r}_a := \langle \mu^*, X_a \rangle$, $q_a = \mathcal{F}_a(r_a)$ be the CDF value of $r_a$ and $\mathcal{F}_a^{-1}$ be the quantile function, i.e. such that $\mathcal{F}_a^{-1}(x) = \inf\{y \in \mathbb{R} : x \leq \mathcal{F}_a(y)\}$. For $\epsilon > 0$ consider the set $E^\epsilon := E_1^\epsilon \cup E_2^\epsilon$ where

$$E_1^\epsilon := \{(x, y) \in [0, 1]^2 : x > y, \mathcal{F}_1^{-1}(x) < \mathcal{F}_2^{-1}(y) - \epsilon\} \ ,$$
$$E_2^\epsilon := \{(x, y) \in [0, 1]^2 : x < y, \mathcal{F}_1^{-1}(x) > \mathcal{F}_2^{-1}(y) + \epsilon\} \ .$$

Note that we can write

$$E_1^\epsilon = \{(x, y) \in [0, 1]^2 : y < x < \mathcal{F}_1(\mathcal{F}_2^{-1}(y) - \epsilon)\} \ ,$$
$$E_2^\epsilon = \{(x, y) \in [0, 1]^2 : \mathcal{F}_1(\mathcal{F}_2^{-1}(y) + \epsilon) < x < y\} \ .$$

Now, let $g_{1,2}(y, \epsilon) = \mathcal{F}_1(\mathcal{F}_2^{-1}(y) + \epsilon)$. Since from Assumption A(ii)(iv), $q_1$ an $q_2$ are $i.i.d$ uniform on $[0, 1]$ we have that

$$
\begin{aligned}
\mathbb{P}((q_1, q_2) \in E^\epsilon) &= \mathbb{P}((q_1, q_2) \in E_1^\epsilon) + \mathbb{P}((q_1, q_2) \in E_2^\epsilon) \\
&= \int_0^1 \int_y^{g_{1,2}(y, -\epsilon)} \mathrm{d}x\mathrm{d}y + \int_0^1 \int_{g_{1,2}(y, \epsilon)}^y \mathrm{d}x\mathrm{d}y \\
&= \int_0^1 \max(g_{1,2}(y, -\epsilon) - y, 0)\mathrm{d}y + \int_0^1 \max(y - g_{1,2}^\epsilon(y, \epsilon), 0)\mathrm{d}y \\
&= \int_0^1 \left[ \max(\mathcal{F}_1(\mathcal{F}_2^{-1}(y) - \epsilon) - y, 0) + \max(y - \mathcal{F}_1(\mathcal{F}_2^{-1}(y) + \epsilon), 0) \right] \mathrm{d}y \ .
\end{aligned}
$$
(21)

Since $\mathcal{F}_1 \neq \mathcal{F}_2$, and $\mathcal{F}_1, \mathcal{F}_2$ are absolutely continuous, there exists $\epsilon' > 0$, such that $\mathcal{F}_2^{-1}(y) - \mathcal{F}_1^{-1}(y) > \epsilon'$, or $\mathcal{F}_2^{-1}(y) - \mathcal{F}_1^{-1}(y) < \epsilon'$ for y inside a closed interval, and hence $\mathbb{P}((q_1, q_2) \in E^{\epsilon'}) > 0$. This yields (20) by letting $\epsilon = \epsilon'$, and $p = \mathbb{P}((q_1, q_2) \in E^\epsilon)$.

Now, let $q_{t,a} = \mathcal{F}_a(\bar{r}_{t,a})$, then for the expected value of the instantaneous standard regret, at round $t$, we can write

$$
\mathbb{E}\left[\bar{r}_{t, a_t^{\mathrm{opt}}} - \bar{r}_{t, a_t^*}\right] \geq \int_{(x,y) \in E^\epsilon} |\mathcal{F}_2^{-1}(y) - \mathcal{F}_1^{-1}(x)| \, \mathrm{d}x \, \mathrm{d}y \geq \epsilon \mathbb{P}((q_{t,1}, q_{t,2}) \in E^\epsilon) = \epsilon p > 0 \ ,
$$

and for the standard regret, we have

$$
\sum_{t=1}^T \mathbb{E}\left[\bar{r}_{t, a_t^{\mathrm{opt}}} - \bar{r}_{t, a_t^*}\right] \geq T \cdot \epsilon p > 0 \ .
$$

Finally, let $\Omega$ be the event that $\frac{1}{2} \cdot \sum_{t=1}^T \mathbb{E}\left[\bar{r}_{t, a_t^{\mathrm{opt}}} - \bar{r}_{t, a_t^*}\right] \leq \sum_{t=1}^T [\bar{r}_{t, a_t^{\mathrm{opt}}} - \bar{r}_{t, a_t^*}]$. Considering the fact that $\sum_{t=1}^T [\bar{r}_{t, a_t^{\mathrm{opt}}} - \bar{r}_{t, a_t^*}] \leq 2L\|\mu^*\|T$, we deduce

$$
\sum_{t=1}^T \mathbb{E}[r_{a_t^{\mathrm{opt}}} - \bar{r}_{t, a_t^*}] = \sum_{t=1}^T \left[ \mathbb{E}[\bar{r}_{t, a_t^{\mathrm{opt}}} - \bar{r}_{t, a_t^*} \mid \Omega]\mathbb{P}(\Omega) + \mathbb{E}[\bar{r}_{t, a_t^{\mathrm{opt}}} - \bar{r}_{t, a_t^*} \mid \Omega^c]\mathbb{P}(\Omega^c) \right]
$$

$$
\leq 2L\|\mu^*\|T\mathbb{P}(\Omega) + \sum_{t=1}^T \mathbb{E}[\bar{r}_{t, a_t^{\mathrm{opt}}} - \bar{r}_{t, a_t^*}]/2 \ ,
$$

and we get $\frac{\epsilon p}{4L\|\mu^*\|} \leq \sum_{t=1}^T \mathbb{E}[\bar{r}_{t, a_t^{\mathrm{opt}}} - \bar{r}_{t, a_t^*}]/(4L\|\mu^*\|T) \leq \mathbb{P}(\Omega)$, which finishes the proof. $\square$

**Remark F.1.** *In Thm. 3, $\epsilon \leq 2L\|\mu^*\|$, otherwise $p = 0$. On the other hand, by the definition $p \leq 1$, and accordingly $\frac{\epsilon p}{4L\|\mu^*\|} \leq 1/2$.*

**Remark F.2.** *With similar reasoning as in the proof of Thm. 3, we can show that if $\mathcal{F}_1 \neq \mathcal{F}_2$ the optimal policy (see Remark 3.1) has linear fair pseudo-regret with positive probability, that is independent of $T$. In particular, there exist $c, c' > 0$, such that for any $T > 0$, $\mathbb{P}(T \cdot c' \leq \sum_{t=1}^T [\mathcal{F}_{a_t^*}(\bar{r}_{t, a_t^*}) - \mathcal{F}_{a_t^{opt}}(\bar{r}_{t, a_t^{opt}})]) > c$.*

**Example 1** (Disjoint supports). *As an example consider the case when $K = 2$ and $\bar{r}_{t,1} - \bar{r}_{t,2} \geq \epsilon > 0$, for all $t \geq 1$, almost surely. Then, $\mathcal{F}_1(\mathcal{F}_2^{-1}(y) - \epsilon) = \mathcal{F}_1(\mathcal{F}_2^{-1}(y) + \epsilon) = 0$ for every $y \in [0, 1]$. Hence we have*

$$
p = \int_0^1 \left[ \max(\mathcal{F}_1(\mathcal{F}_2^{-1}(y) - \epsilon) - y, 0) + \max(y - \mathcal{F}_1(\mathcal{F}_2^{-1}(y) + \epsilon), 0) \right] \mathrm{d}y = 1/2 \ .
$$

*Then by Thm. 3, with probability at least $\frac{\epsilon}{8L\|\mu^*\|}$, for any $T > 0$, we have $\sum_{t=1}^T [\bar{r}_{t, a_t^{opt}} - \bar{r}_{t, a_t^*}] \geq \frac{T\epsilon}{4}$.*