# OpenReview forum: "Group Meritocratic Fairness in Linear Contextual Bandits"
_NeurIPS.cc/2022/Conference — NeurIPS 2022 Accept_

### Official Review · Reviewer_4BQR · 2022-06-22

**Rating:** 6
**Confidence:** 4
**Soundness:** 3 good
**Presentation:** 4 excellent
**Contribution:** 3 good

**Summary:**

The paper addresses a bandit problem with fairness towards sensitive groups. Precisely, the formal problem studied is a linear contextual bandit problem where each arm represents a group. The notion of group meritocratic fairness consists in selecting candidates with highest relative rank within their group. In the bandit setting, this requires to estimate the cumulative distribution function (CDF) of the rewards. A bandit algorithm is then evaluated by the fair pseudo-regret, which is calculated based on the CDFs rather than the rewards in the standard pseudo-regret. The paper presents a linear bandit algorithm, Fair-Greedy, and provides a theoretical guarantee on its fair pseudo-regret. Its performance is evaluated on synthetic environments.

**Questions:**

Could you precisely describe a scenario which maps to the formal problem studied in the paper, as a response to (W1)? Could you also respond to the concerns raised in (W2) and (W3)?

**Limitations:**

Limitations are discussed in Section 7.

**Strengths And Weaknesses:**

Strengths:

S1. The paper presents a new bandit problem, with the specific theoretical challenge of estimating the relative rank of an arm, which requires learning the CDF of the rewards

S2. The results presented in the paper are technically sound, and the steps / lemmas towards the main result on fair regret (Theorem 1) are clearly presented, and of interest.

S3. The meritocratic fairness criterion, derived from Kearns et al. [22], is itself valuable and only received little attention in the fair machine learning literature. Extending the idea of comparing individuals within their sensitive group to sequential decision-making is a promising research direction, because hiring processes are often of sequential nature. (However, the chosen formulation is not convincing, see below).

Weaknesses: (W1) Unrealistic setting, (W2) No theoretical guarantee on standard regret, (W3) Lack of empirical comparison with existing fair bandit algorithms

W1. The formal setting studied in the paper is not realistic. In the proposed formulation, each arm corresponds to a sensitive group. Concretely, this means that at each round, the decision-maker must choose between one man and one woman, which is a dubious scenario (this is the example used in the experiments on US Census Data in Appendix D.2). Furthermore, in such selection/hiring problems, fairness concerns also arise because of the presence of a majority group with many applicants and a minority group with few (e.g. gender and racial imbalance in applications for tech jobs or computer science degrees) – this hardly fits in the current framework where one group = one arm.

The authors make the assumption that the rewards of each arm are mutually independent (Assumption A-iii), which is only reasonable if arms indeed represent distinct groups. Aware of this limitation, the authors suggest (L191) a relaxation where groups can be represented by more than one arm. This is not convincing because A-iii still requires the context-arms $(X_a)_a$ to be independent, so the relaxation only holds when arms do not provide context.

Overall, I think there is a gap between the informal description of the problem which is very reasonable (S3), and the bandit formulation which is very far from modelling a real-world scenario. The experiments, either on toy environments or the US Census Data, do not support the proposed setting.

W2. What are the theoretical guarantees of the algorithm in terms of standard regret? Without such result, it is difficult to evaluate the cost of fairness. The simulations in Section 5 suggest there is a trade-off between standard regret and fair regret. The authors do mention in Remark 3.1 that fair regret and standard regret are in competition but only analyse the standard regret in the edge case where all arms have the same reward distribution.

W3. Empirically, how do the proposed algorithms compare to other fair bandit algorithms? e.g. Patil et al [31], who aim to balance between selecting high quality individuals while guaranteeing every arm a minimal probability of being selected. This is important because the authors study a similar guarantee of *demographic parity* for their Fair-Greedy algorithm (Proposition 5.1, L286-287).

---

> ### Author Response · Authors · 2022-08-02
> **Response to Review 4BQR**
>
> We thank the reviewer for the useful feedback and address their concerns below.
>
> **One arm = one group restriction not so realistic (W1).**
> We agree that this restriction is a simplifying assumption and not fully applicable to the hiring scenarios considered in the paper. However, we think our work can be of interest even with this assumption as it simplifies the problem and notation while retaining novelty of the setting and technical challenges which, to the best of our knowledge, we are the first to address. We also think that our suggested extension, where more than one arm represents the same sensitive group, is one step forward to a more realistic setting and does not  change the theoretical analysis. Regarding this extension, the reviewer wrote that, since we have independence of contexts across arms, the only solution is that no context is provided. *We disagree with this statement:* we can still have contexts, they just will be sampled from the same distribution (that of the group) as in the case when some of the CDFs ($\mathcal{F}_a$) are equal, this however, doesn’t mean they do not contain information on what arm to choose. In case we did not understand the reviewer comment or if our answer needs more clarifications, we’d be glad to discuss this further.
>
> Despite the above discussion, this suggested extension still does not capture the scenario where the number of contexts for each group varies across rounds. For this reason we updated the manuscript to include in Appendix E a natural extension of our original setup, where we can easily adapt our regret results. In the following, we provide an overview.
>
> We considered the following setting. At each round $t$ the agent receives $\\{(X_{t,a}, s_{t,a})\\}\_{a}^K$ where $s_{t,a} \in [G] $ is a random variable indicating the sensitive group of the context $X_{t,a}$ and after selecting an action the agent receives the noisy reward $\mu^{\star,\top}X_{t,a} + \eta_t$. This setting is indeed more realistic as it includes for instance the case where at each round $\\{(X_{t,a}, s_{t,a})\\}\_{a}^K$ are i.i.d. from the underlying population and the number of contexts in a group at each round reflects the true percentage of candidates from that group (note that $X_{t,a}$ and $s_{t,a}$ can still be dependent).
> Assuming that $\\{(X_{t,a}, s_{t,a})\\}\_{t=1}^T$ are i.i.d. copies of the random variable $(X_a, s_a)$, we can then redefine the relative rank of $X_{t,a}$ as $\mathcal{F}(\mu^{\star\top}X_{t,a} , s_{t,a})$ with $\mathcal{F}(r, i) = \mathbb{P}(\mu^{*\top}{X_a}  \leq r | s_a = i)$ and naturally extend the definition of GMF policy, the fair-regret, the assumptions, and our fair-greedy policy to this more general setting. After this, we can recover similar high probability bounds for the fair regret with Theorem 1 as a special case when $K=G$ and $s_a=a$. Note that the analysis of this extension is quite straightforward since the same proof techniques are used with the only addition of a Chernoff bound to lower bound the number of contexts used for the approximate CDFs at round $t$, which is now a random variable.
> Interestingly, when  $\\{(X_{t,a}, s_{t,a})\\}\_{a}^K$ are i.i.d., this setup improves the dependency on the number of arms when $K > G^{3/2}$, e.g. the $K^3$ term in the bound of Theorem 1 becomes $K^2G^{3/2}$. This happens because in this case, to approximate the CDF of each group, we can use contexts coming from more than one arm.
>
>
>
> **Guarantees for standard regret and tradeoff between fair and standard regret (W2).**
> We argue that fair-regret can substitute standard regret in the case where rewards are incomparable between groups. However, studying the tradeoff is important and we added appendix F and some comments in Remark 3.1 regarding this. Please see also the first point of the general response for more details.
>
>
> **Empirical comparison with other fair-bandit algorithms e.g. Patil et al. … demographic parity (W3).**
> The method by Patil et al. 2020 is a method designed for bandits without contexts and hence it should first be adapted to include context information before any comparison.
> Please see the second point of the general response for more details.

---

> > ### Comment · Reviewer_4BQR · 2022-08-08
> > **Thank you for your detailed response**
> >
> > I really appreciated the detailed response provided by the authors and the important additions made to the paper, which effectively address my main concern about the setting "one arm = one group." In the previous version of the paper, I indeed understood that independence of contexts would lead to equal $\mathcal{F}_a$, which is not an interesting setting because fairness issues arise when groups have different reward distributions (e.g., because of historical or measurement bias). Yet the extension of the framework which has been added in Appendix E provides a more realistic and acceptable treatment of sensitive groups. I also appreciate the additional experiments which support the effectiveness of the algorithm in this setting.
> >
> > Regarding my (and other reviewers') question on standard vs. fair regret, I also find the additional comments in Remark 3.1 and Appendix F useful. I am increasing my score.

---

> > > ### Author Response · Authors · 2022-08-08
> > > **Thank you!**
> > >
> > > Many thanks for increasing the score and for your valuable feedback!

---

> ### Author Response · Authors · 2022-08-08
> **Follow up Letter to Reviewer 4BQR**
>
> Dear Reviewer 4BQR,
>
> based on your concerns we have added the following results to the paper:
>
> - We address both theoretically and experimentally the issue of the “one arm = one group” restriction (weakness W1 in your review), showing that our results can be easily adapted to the more general scenario (see Appendix E).
> - We address the tradeoff between fairness and reward maximisation (weakness W2 in your review) by proving that, when $K=2$ and the reward distributions are absolutely continuous, independent but NOT identically distributed our policy achieves linear standard regret with positive probability (see the updated Remark 3.1 and Appendix F).
>
> As the response period ends tomorrow **we would appreciate your feedback on our revisions**, which we believe mitigate your concerns. In particular, we are eager to know if you agree with the above sentiment and would be willing to reevaluate the paper accordingly. We really appreciate your feedback and suggested improvements.

---

### Official Review · Reviewer_kA7t · 2022-07-11

**Rating:** 5
**Confidence:** 3
**Soundness:** 3 good
**Presentation:** 2 fair
**Contribution:** 2 fair

**Summary:**

This paper considers group fairness in linear contextual bandits. Specifically, the paper proposes a notion of fairness trying to capture `` given an arriving individual, how good the reward is when compared to candidates from the same group''. The paper also proposes a fairness-related regret. Then the paper proposes a Fair-Greedy policy that provably reaches sub-linear regret. Numerical simulation provided.

**Questions:**

1. Problem formulation is confusing in section~3.
* Line 125, "this objective might be unfair to some of the sensitive groups". I would suggest explaining a bit about why the classic objective is unfair.
* Line 127, "i.i.d. copies of the random variable $X_a$". I don't quite understand what is an "i.i.d. copy of a random variable", and I don't see a definition of r.v. $X_a$.
* Line 130, "a reward sampled from the distribution of arm $a$". I don't see a definition of the distribution of arm $a$.

For me the problem formulation seems confusing, and some sentences also deliver unclear or ambiguous meaning, e.g., Line 130. I had difficulty understanding the problem setting, and I was lost when reading Definition 3.1. Please try to express ideas accurate and clear,
 and give more explanation on complex notations.

2. The definition of regret. In Remark 3.1, the authors states that the proposed regret is similar to the standard pseudo-regret definition, which I cannot see through based on the given formulation. It would be helpful if the authors could provide more discussions or even proofs if necessary. Moreover, in general cases, fairness and regret minimization can be either opposite goals or aligned goals, which depends on their definitions. So, the statement in Line 148 is one-sided.

**Limitations:**

1. The paper needs further polishing on writing especially in section 3, to better deliver the ideas.

2. Fairness notion. If my guess is correct, the relative rank $\mathcal{F}_a(\cdot)$ is defined to measure how good an individual is on generating reward comparing to the other individuals in the same sensitive group. If that is the case, then the proposed fairness notion aligns with reward maximization, and this might be able to explain my second question in Questions section. However, in this formulation, any existing policy on linear contextual bandits with $O(\sqrt{T})$ regret will achieve the same fairness objective proposed in this paper, with only constant difference. In other words, the proposed fairness constraint equals the goal of reward maximization. Thus, the problem goes back to a standard linear contextual bandits problem, which is of less interest. This is one of my major concerns of the paper.
To highlight the contributions of this paper if any, I would suggest highlighting the conflict of simultaneously satisfying fairness constraint and maximizing reward, or explicitly showing how much tradeoff has been made to achieve fairness constraint.

**Strengths And Weaknesses:**

Strengths:
* The problem is practical and can be applied to many real-world applications.
* Related work is comprehensive and well discussed. This paper tackles a problem that is technically non-trivial.

Weaknesses:
* The writing can be further polished.
* The regret definition and fairness notion might be too aligned, that mitigates the fairness constraint in learning process.

---

> ### Author Response · Authors · 2022-08-02
> **Response to Reviewer kA7t**
>
> We thank the reviewer for the useful feedback and address their concerns below.
>
> **Writing should be polished, especially Section 3. I do not understand the fairness definition.**
> We believe the paper to be sufficiently well-written and think that the positive response about clarity from reviewers subD and 4BQR corroborate this. The notation is not overly complex and suitable for what it is set out to achieve. However, to fix the confusion over the problem formulation in section 3, we have clarified each bullet point in the revised version. For example: *What are i.i.d. copies?* “i.i.d. copies of $X_a$” is a shorthand for “i.i.d. and have the same distribution as $X_a$”, which is also where we define that random variable; we clarified this point. *How are standard regret and fair regret similar?* The definitions resemble each other and we made this more precise in Remark 3.1. *Fairness and reward maximization can be aligned or not. The statement at the end of remark 3.1 is one sided since it considers only the case where it aligns* We added to the remark an example where fairness and reward maximization do not align to clarify this aspect. We hope these clarifications improve the readability and we would be happy to answer any other questions from the reviewer regarding the presentation of the paper.
>
>
> **The proposed fairness notion aligns with reward maximization… A policy achieving $O(\sqrt(t))$ standard regret also achieves the same fair regret.**
> This is false. On the contrary, whenever one of the CDFs of the arms is different than that of the others, the objectives are competing. In particular, as we now explain in Remark 3.1, when rewards are absolutely continuous and independent across arms, with one arm always dominating the others in terms of reward, with non-zero probability, the GMF policy has linear standard regret, while the optimal policy has linear fair regret. See also the first point in the general response for a more in depth discussion. We’d be glad to know if the reviewer agrees on this point or if further clarification is needed.

---

> > ### Comment · Reviewer_kA7t · 2022-08-08
> > **Rebuttal Reply**
> >
> > Thanks for the explanations. The updated version looks clearer on problem formulation, and the updated Remark~3.1 makes more sense. Now I am good with the fairness notion and the regret definition. Thus, I will increase my rating to a borderline accept.

---

> > > ### Author Response · Authors · 2022-08-08
> > > **Thanks for increasing the score!**
> > >
> > > We would like to ask you if your previous concerns about *“Presentation”* and especially  *“Contribution”* have been resolved (we notice that the score for those remained unchanged). We’d be eager to address any final comments you may have in order to
> > > further improve your consideration of our paper.
> > >
> > > Thank you!

---

### Official Review · Reviewer_subD · 2022-07-11

**Rating:** 7
**Confidence:** 4
**Soundness:** 3 good
**Presentation:** 3 good
**Contribution:** 3 good

**Summary:**

This work studies an important problem class of fairness-aware decision making on how to select candidate from pools without scarify the rights of candidates from sensitive groups.  The approach is to work on the notion of group meritocratic fairness, to prioritize the resources to the candidates with highest relative in-group ranking. The authors propose the Fair-Greedy policy (Algorithm 1) with demographic parity property (Lemma 4.1). The key to unlock regret analysis is to allow arguments in classical stochastic linear bandit in bounding relative rank for  instance regret bound (Lemma 5.2) . Simulation studies observe that the Fair-greedy tradeoff pseudo fair regret with standard pseudo regret.


**Questions:**


1. **Insights on exploration for maximizing Group Meritocratic Fairness**.  As you may already aware, Greedy-style bandit policy doesn't require us to think deep in how to "explore to maximize the objective"--the greedy policy just impute the estimate and take what seems best. Such policy style ignores the potential of using estimate uncertainty to improve exploration efficiency. Therefore, my question would be, from your perspective, how to use estimate uncertainty to design a bandit policy for maximizing Group Meritocratic Fairness? I would be my great appreciation if the authors can share their insights on this regard.

**Limitations:**


1. I think the main limitations is the lack of fairness-aware baselines. As a reader, I would expect a more inviting comparisons of the proposed methods with other existing fairness-aware bandit algorithms. But this is not a drawback of the paper, and I think the authors have done very good job on convincing readers about the advantage of their Fair-Greedy policy for maximizing Group Meritocratic Fairness.

**Strengths And Weaknesses:**

Strengths:
1. The current submission is written clear, elegant and easy to follow.
2. The intent to bridge fairness metric into bandit community is highly welcomed in my opinion.

Weaknesses:
1. Baseline should include fairness-aware bandit algorithms. I appreciate the insights on comparing Fair-Greedy with OFUL and Uniform Random, but it would be more informative if the authors provide comparison with fairness-aware bandit algorithms.

---

> ### Author Response · Authors · 2022-08-02
> **Response to Reviewer subD**
>
> We thank the reviewer for the useful feedback and their appreciation of our work. We  address their concerns below.
>
> **Baseline should include fairness-aware bandit algorithms.**
> As there are no fairness-aware algorithms, except for our fair-greedy policy, achieving sub-linear GMF regret, we think this is unnecessary. See also the second point in the general response for more details.
>
> **Insights on exploration for maximizing Group Meritocratic Fairness.**
> Using confidence intervals is not useful under our assumptions since the GMF policy, as the fair-greedy policy, always explores all arms. See also the third point of the general response for more details.

---

### Official Review · Reviewer_XPC5 · 2022-07-11

**Rating:** 7
**Confidence:** 4
**Soundness:** 3 good
**Presentation:** 3 good
**Contribution:** 3 good

**Summary:**

The paper studies a contextual bandit problem where different arms correspond to different populations. The goal of the paper is to simultaneously ensure low regret *and* fairness. Here the definition of fairness aims to guarantee that candidates with higher rank are selected over candidates of lower rank. Here, the rank of agent t is the candidate’s probability to be better than a randomly selected candidate, or in other words the fraction of candidates in the distribution that t is better than.

**Questions:**

- Can the authors provide more intuition and explanation of Assumption iv?
- Can the authors say a word about relaxing the assumption that \mu^* is the same in all groups?
- Can the authors say a word about exploring as they exploit rather than having first an explore then an exploit phase, and can this help reduce the dependencies in K?


**Limitations:**

Limitations on the algorithm and assumptions are adequately discussed. N/A with respect to negative societal impact, as the entire point of the paper is to avoid such negative impact.

**Strengths And Weaknesses:**

Strengths:
- The paper provides an algorithm that has sub-linear (\sqrt(T)) fair pseudo-regret. This is nice because this definition of regret requires that the individuals selected in each round have overall high rank within their group, which both guarantees good performance (as we are hiring the best individuals) and fairness within each group (as we favor individuals of higher rank over individuals of lower rank)
- The algorithm also guarantees some form of fairness across groups, in that it does not over-select candidates from a subset of the arms. Rather, one of the main lemmas/technical contributions of the paper is to show that each arm is sampled uniformly at random.
- I think the main technical trick here is simple and neat: because the rank of an agent is the cdf of the reward distribution, the distribution of rank is uniform and identical across different groups! In turn, the probability of arm a having the highest-rank candidate in each round is also 1/K. This allows the algorithm to simultaneously pick a candidate that is with high probability good/does not add to the regret, while at the same time incentivizing exploration of all arms.

Weaknesses:
- I think the definition of fairness could be stronger while keeping the paper tractable. Here, the definition of fairness does not make direct comparisons across different groups, other than through the relative rank in each group. However, i) someone with high relative rank in group 1 might be much more or much less qualified than someone with the same relative rank in group 2, depending on the distribution of contexts for each population, which is not taken into account in the paper; ii) I believe that the global, non-group-specific rank can be understood here since the algorithm in the paper requires estimating \mu^* and the distribution of contexts in each group, which gives enough information to compute the rank across groups
- While the distributions of contexts/attributes are different in different populations, the parameter \mu^*/the mapping from contexts to rewards/scores is the same across different groups. I think one of the main difficulties in the space of fairness is that in general, the same attributes have different predictive power across different populations (ex: high SAT scores mean different things if you have access to prep and can take the SAT several times and report only your best score, versus if you cannot afford prep and can only take the SAT once).
- I believe the current approach is an explore-exploit type of approach where first, the algorithm estimates \mu^*, then it exploits. However, one thing that the algorithm could do is to exploit the entire time based on the currently best known estimate \mu_t of \mu^*; the properties of the current algorithm that each arm would be explored with probability 1/K would be still true, and for any round t we would have a \sqrt{t}-accurate estimate of  \mu^* and a K \sqrt{t}-accurate estimate of the cdf for each arm. This hopefully could hopefully improve on the K^3 dependency in the regret bound.
- Assumptions i, ii, and iii seem fairly reasonable and classical. However, assumption iv is hard to parse; I think it would be useful to have more discussion of what this assumption actually means, how strong it is, and example of classical distribution that satisfies and do not satisfy this assumption.

I think the paper is interesting and I think it would be nice to have it as part of the NeurIPS program. I’m currently weakly recommending acceptance, where the “weakly” is due to the fact I believe i) the regret bounds can be improved and ii) I think it might be possible to take further fairness considerations into account in the approach of the paper.

---

> ### Author Response · Authors · 2022-08-02
> **Response to Reviewer XPC5**
>
> We thank the reviewer for the useful positive feedback and address their concerns below.
>
> **I think the definition of fairness could be stronger while keeping the paper tractable… compute global rank.**
> We note that since in the analysis we have rewards mutually independent across arms and absolutely continuous, at round $t$, while rewards and relative ranks from different arms can be similar, they are equal with zero probability. When relative ranks are similar it can make sense to pick the arm with the best reward, achieving a tradeoff between the two, also because, as the reviewer mentions, it is estimated by the fair-greedy policy to estimate the relative rank. However, we believe this to be outside the scope of this work. See also the first point of the general response which relates to this aspect.
>
> **I believe the current approach is an explore-exploit type of approach**
> *Our approach is greedy from the start: no exploration phase is needed.* Please see the third point of the general response for more details. The $K^3$ term in the bound can get arbitrarily close to $K^2$ by increasing the constants, however at the moment we don’t have an idea on how to improve it further.
>
> **Can the authors provide more intuition and explanation of Assumption (iv)?**
> We added the following clarifying bit in the updated draft after the assumptions. *“Assumption A(iv) implies that $\mu^{\star \top}X_a$ is absolutely continuous and is satisfied when $X_a$ is absolutely continuous in a subspace of $\mathbb{R}^d$ which is not orthogonal to $\mu^\star$"*. We also added *“It does not cover cases where, e.g. $X_a$ is a sum of two independent and absolutely continuous random variables in orthogonal subspaces of $\mathbb{R}^d$”*.   We describe the flexibility of this assumption in the next point.
>
> **parameter $\mu^\star$, the mapping from contexts to rewards/scores, is the same across different groups.**
> We note that Assumption A(iv) is actually flexible enough to allow $\mu^\star$ to model group-specific effects, effectively capturing the case where there is a different vector $\mu^\star_a$ for each group. We can see this when for example $\mu^\star \in \mathbb{R}^d$, $d=2K$ and $X_a$ is an absolutely continuous random variable in a 2-dimensional subspace of $\mathbb{R}^d$ involving just 2 coordinates, that are different for each group $a \in [K]$. In this way only the corresponding two coordinates of $\mu^\star$ will be used (exclusively) by arm $a$. We note that this is also the case in the simulation experiment that we present in Section 6.

---

> > ### Comment · Reviewer_XPC5 · 2022-08-07
> > **Thanks for the response!**
> >
> > Hi and thanks for the response!
> >
> > With respect to i), I agree this might be left to future work, though I am now very curious about this extension.
> > ii) I am still a bit confused about this point, I thought the first phase was mostly used to compute $\mu_{t_0}$. Am I missing something here/how does the exploitation work before $t_0$?
> > iii) Thanks for the explanation! It does clear up things a bit and I find the added explanation useful.
> > iv) Thanks for clarifying this! This one of my main worries reading the paper, and this seems addressed.
> > Given that my main concerns are addressed, I will increase my score to a 7. I also just wanted to note that I really appreciate and am impressed by the amount of effort that the authors put in their response/revision/new results. Cheers!

---

> > > ### Author Response · Authors · 2022-08-07
> > > **Thank you for increasing the score and appreciating our effort!**
> > >
> > > Here is the response to point ii).
> > >
> > > **ii) “I thought the first phase was mostly used to compute $\mu_{t_0}$”**
> > > The phase the reviewer mentions happens every round $t \geq 3$. In particular, at every round $t$ we use the information (context and rewards) collected in the first $t_0 = \lfloor(t-1)/2\rfloor$ rounds to construct $\mu_{t_0}$, then we use the information in the remaining rounds to approximate the relative rank. Finally, we use a greedy approach by selecting the arm/context with the highest approximate relative rank. Thus, there is no exploration phase and $\mu_{t_0}$ is computed at every round. We believe the confusion stems from the fact that the dependence of $t_0$ on $t$ is not explicit. We will make this point clearer in the updated version of the paper.
> > >
> > > If there are still unclear points we’d be glad to discuss further.

---

> > > > ### Comment · Reviewer_XPC5 · 2022-08-08
> > > > **Thanks for the clarification**
> > > >
> > > > Thanks! It'd be great to make this dependence explicit in the algorithm, agreed

---

### Author Response · Authors · 2022-08-02
**General response to all reviewers.**


We thank the reviewers for their thoughtful feedback. We uploaded a revision of the paper (with major changes in blue) containing minor corrections and some improvements based on the feedback of the reviewers. In particular, we highlight the following additional results:

1. **We studied standard regret guarantees for the group meritocratic fair (GMF) policy** (see point 1 below), proving that it achieves linear regret with non-zero probability when the distributions differ between groups. This means that in most cases our fairness notion and reward maximization are competing objectives.
2.  **We extended our setting removing the restrictive “one group = one arm” assumption** and provided similar regret guarantees with a straightforward extension of the analysis (see answer to W1 of 4BQR).
3. **We provided an experiment with US census data where at each round 10 candidates are sampled i.i.d. together with their sensitive group**, which is the ethnicity in this case (In Appendix E.1). This experiment fits in the extension studied at point 2 and provides a more realistic scenario. Results show that our policy achieves good fair regret also in this case and that it selects more candidates from  underperforming minority groups compared to policies maximizing the standard regret.

Overall these changes did not affect the results already present in the paper and its structure. The additional results are added to the appendix and mentioned in the main body of the paper.

We address the reviewers’ major common points (In bold in the comment below) and also answer each reviewer individually. We’d be glad to answer/address any further questions/comments by the reviewers.

---

> ### Author Response · Authors · 2022-08-02
> **Response to reviewer's common points**
>
> **1) Guarantees for standard regret and tradeoff between fair and standard regret…** The objective of this work is not to have good standard regret subject to fairness constraints but to *“only” learn a policy with sublinear fair-regret*. We stress that *this can be already the final objective when the rewards from different groups are incomparable*, as in the academic example that we present in the introduction, where we would like to hire candidates for different research subjects and rewards are the number of citations, which varies greatly across subjects. There, it makes little sense to maximize the reward since it is not a good performance measure across groups. In a sense, for this kind of problems, the *fair-regret can substitute the standard regret entirely and no tradeoff between the two is needed*. A similar reasoning (with the same example) is also presented in [1], which introduced the notion of relative rank that we use. Furthermore, we argue in the paper that achieving sublinear fair regret is already a more difficult problem than achieving sublinear standard regret, since the agent has also to estimate the reward distributions. This problem presents unique technical challenges, some new for contextual bandits and some new to the non-interactive setting of [1].
>
> Despite this, we believe the question about the tradeoff between fair and standard regret is worth exploring in problems where comparing rewards makes sense also across groups. For example, when rewards measure the performance on the job, groups represent ethnicities and the agent is an employer who would like to maximize rewards under some government regulation on fairness. We began to study this tradeoff by showing that, *when there are two arms with different reward distributions, the GMF policy has **linear** standard regret with non-zero probability*. We also characterize this probability for the case where the support of the reward distributions are disjoint. The proof is based on the fact that if at each round, rewards are sampled independently between arms and the reward distributions are continuous and different, then cases where the optimal policy and the GMF policy select different arms happen with non-zero probability. We refer to Remark 3.1 and Appendix F of the updated manuscript for more details and rigorous proofs.
>
>
> **2) Empirical comparison with other fair-bandit algorithms… demographic parity**
> We remark that, to the best of our knowledge, our notion of fairness based on relative ranks has never been studied before in the bandit setting. The fact that the GMF fair policy and our fair-greedy policy also satisfy some notions of demographic parity is a consequence of the setup that we study and does not directly align with the notion of group meritocratic fairness. Comparisons with other fairness-aware algorithms (FAAs) become less insightful since they use different notions of fairness. In addition, the method suggested by 4BQR (in [2]) is a method for bandits without contexts and hence it should first be adapted to include context information before any comparison.
>
> **3) Explore-exploit approach might be suboptimal. Could just exploit or use confidence intervals.**
> *As noted in the paper, our policy is greedy from the start (no exploration phase), although it still guarantees sufficient exploration and consequently achieves sublinear regret.*  This is because in our setting rewards are absolutely continuous random variables and independent across arms. This ensures that the (true) relative ranks are i.i.d uniform on [0,1] at each round and consequently the GMF policy selects each arm with equal probability. The fact that we require $t$ to be of $\tilde{\Omega}(K^{3})$ to bound the instantaneous regret is a consequence of our analysis and does not come from an exploration phase. Furthermore, since the greedy policy works well and explores all arms, there is no need to use confidence intervals, whose goal is precisely to avoid that important arms go unchosen. However, an interesting future research direction could be to remove the assumption that the contexts are independent across arms. In this case, the greedy approach might not work and confidence intervals may be helpful together with novel analysis.
>
>
> References
>
> [1] Kearns, Michael, Aaron Roth, and Zhiwei Steven Wu. "Meritocratic fairness for cross-population selection." International Conference on Machine Learning. PMLR, 2017.
>
> [2] Patil, Vishakha, et al. "Achieving Fairness in the Stochastic Multi-Armed Bandit Problem." J. Mach. Learn. Res. 22 (2021): 174-1.

---

### Meta-Review · Area_Chair_Pwwx · 2022-08-28

**Recommendation:** Accept
**Confidence:** Certain

**Metareview:**

Reviewers agree on the merits of sharing the paper with the community.The authors are highly encouraged to incorporate the many constructive suggestions offered.

**Award:**

No

---

### Decision · Program_Chairs · 2022-09-14

Accept